# RNF43 inhibits WNT5A-driven signaling and suppresses melanoma invasion and resistance to the targeted therapy

Tomasz Radaszkiewicz[1], Michaela Nosková[1], Kristína Gömöryová[1],
Olga Vondálová Blanářová[1], Katarzyna Anna Radaszkiewicz[1], Markéta Picková[1,2,3],
Ráchel Víchová[2], Tomáš Gybel[4], Karol Kaiser[1], Lucia Demková[4], Lucia Kučerová[4],
Tomáš Bárta[1,5], David Potěšil[6], Zbyněk Zdráhal[6], Karel Souček[1,2,3],
Vítězslav Bryja[1,2]*

[1]Department of Experimental Biology, Faculty of Science, Masaryk University,
Brno, Czech Republic; [2]Department of Cytokinetics, Institute of Biophysics CAS,
Brno, Czech Republic; [3]International Clinical Research Center FNUSA-ICRC, Brno,
Czech Republic; [4]Laboratory of Molecular Oncology, Cancer Research Institute,
Biomedical Research Center of the Slovak Academy of Sciences, Bratislava, Slovakia;
[5]Department of Histology and Embryology, Faculty of Medicine, Masaryk University,
Brno, Czech Republic; [6]Central European Institute of Technology, Masaryk University,
Brno, Czech Republic

**Abstract** RNF43 is an E3 ubiquitin ligase and known negative regulator of WNT/β-catenin signaling. We demonstrate that RNF43 is also a regulator of noncanonical WNT5A-induced signaling in human cells. Analysis of the RNF43 interactome using BioID and immunoprecipitation showed that RNF43 can interact with the core receptor complex components dedicated to the noncanonical Wnt pathway such as ROR1, ROR2, VANGL1, and VANGL2. RNF43 triggers VANGL2 ubiquitination and proteasomal degradation and clathrin-dependent internalization of ROR1 receptor and inhibits ROR2 activation. These activities of RNF43 are physiologically relevant and block pro-metastatic WNT5A signaling in melanoma. RNF43 inhibits responses to WNT5A, which results in the suppression of invasive properties of melanoma cells. Furthermore, RNF43 prevented WNT5A-assisted development of resistance to BRAF V600E and MEK inhibitors. Next, RNF43 acted as melanoma suppressor and improved response to targeted therapies in vivo. In line with these findings, *RNF43* expression decreases during melanoma progression and RNF43-low patients have a worse prognosis. We conclude that RNF43 is a newly discovered negative regulator of WNT5A-mediated biological responses that desensitizes cells to WNT5A.

*For correspondence:
bryja@sci.muni.cz

Competing interest: The authors declare that no competing interests exist.

## Introduction

Ubiquitination is a post-translational modification (PTM) based on the addition of the evolutionary conserved protein ubiquitin (Ub) to the lysine residue(s) of the modified protein (*Hershko and Ciechanover, 1998*). Ubiquitination controls the turnover, activation state, cellular localization, and interactions of target proteins. Undoubtedly, it is a process that has a direct impact on various aspects of cell biology (*Rape, 2018*). Ubiquitination requires sequential activation of ubiquitin, its transfer to the carrier protein, and subsequent linkage reaction with the substrate lysine residues. This last step, mediated by the E3 ubiquitin protein ligases (E3s), determines target specificity.

Ring Finger protein 43 (RNF43) is a E3 ubiquitin ligase with a single transmembrane domain from the PA-TM-RING family. RNF43 and its close homolog Zinc and Ring Finger 3 (ZNRF3) act

as negative regulators of the Wnt/β-catenin signaling pathway (*Koo et al., 2012*; *Hao et al., 2012*). Wnt/β-catenin signaling is an evolutionary conserved pathway and a crucial regulator of embryonal development and tissue homeostasis. RNF43 and ZNRF3 control via regulation of Wnt/β-catenin multiple processes including liver zonation (*Planas-Paz et al., 2016*), limb specification (*Szenker-Ravi et al., 2018*), and mammalian sex determination (*Harris et al., 2018*). Mechanistically, RNF43 and ZNRF3 ubiquitinate plasma membrane Wnt receptors called Frizzleds (FZDs) and a co-receptor low-density lipoprotein receptor-related protein 6 (LRP6), which results in their internalization and degradation (*Hao et al., 2012*; *Koo et al., 2012*). Therefore, cells become less sensitive or insensitive to Wnt ligands. Activity of RNF43/ZNRF43 is regulated by secreted proteins from R-spondin (RSPO) family (*Kazanskaya et al., 2004*; *Kim et al., 2008*; *Kim et al., 2006*; *Kim et al., 2005*; *Nam et al., 2007*; *Nam et al., 2006*; *Peng et al., 2013*; *Xie et al., 2013*) that trigger internationalization of RNF43/ZNRF3 and function as physiologically relevant activators of Wnt/β-catenin pathway (*Binnerts et al., 2007*; *Carmon et al., 2011*; *de Lau et al., 2011*; *Hao et al., 2016*; *Hao et al., 2012*; *Jiang et al., 2015*; *Koo et al., 2012*; *Zebisch et al., 2013*; *Zebisch and Jones, 2015*).

Because deregulation of Wnt/β-catenin pathway promotes tumor formation (*Lim and Nusse, 2013*; *van Kappel and Maurice, 2017*; *Wiese and Nusse, 2018*), RNF43/ZNRF3 can act as tumor suppressors. Indeed, mutation or inactivation of *RNF43/ZNRF3* leads to the oncogenic activation of Wnt signaling and associates with colorectal, liver, gastric, endometrial, ovarian, and pancreatic cancers (*Bond et al., 2016*; *Eto et al., 2018*; *Giannakis et al., 2014*; *Jiang et al., 2013*; *Jo et al., 2015*; *Niu et al., 2015*; *Planas-Paz et al., 2016*; *Ryland et al., 2013*; *Spit et al., 2020*; *Tsukiyama et al., 2020*).

Some members of the Wnt family – such as WNT5A and WNT11 – preferentially activate downstream signaling that is distinct from Wnt/β-catenin pathway and is referred to as β-catenin-independent or noncanonical Wnt pathway (*Pandur et al., 2002*; *Humphries and Mlodzik, 2018*; *VanderVorst et al., 2019*; *Andre et al., 2015*). Noncanonical Wnt pathway shares some features with the Wnt/β-catenin pathway – such as the requirement for FZD receptors, dishevelled (DVL) phosphoprotein and casein kinase 1 (CK1) – but clearly differs in others. In the mammalian noncanonical pathway, receptor tyrosine kinase-like orphan receptor 1 (ROR1) and ROR2 act as primary (co-)receptors (in contrast to LRP5/6 that have this role in the Wnt/β-catenin pathway) and four-transmembrane Vang-like protein 1 (VANGL1) and VANGL2 participate in the signal transduction (*Asem et al., 2016*; *VanderVorst et al., 2019*). This signaling axis is also referred to as planar cell polarity pathway (PCP), and its activation leads to changes in actin cytoskeleton dynamics, facilitating, that is, polarized cell migration (*Andre et al., 2015*; *Janovská and Bryja, 2017*; *Kaucká et al., 2015*; *Weeraratna et al., 2002*).

FZD receptors, the best-defined targets of RNF43/ZNRF3, are shared among all Wnt pathways and their endocytosis and/or degradation have the potential, at least in theory, to prevent signaling by any Wnt ligands. So far, there is only one study that suggests the role of RNF43/ZNRF3 in noncanonical Wnt signaling in mammals (*Tsukiyama et al., 2015*). In addition, secreted inhibitor of RNF43/ZNRF3 called r-spondin 3 (RSPO3) potentiated noncanonical PCP pathway in *Xenopus* in a Wnt5a and dishevelled-dependent manner (*Glinka et al., 2011*; *Ohkawara et al., 2011*). In mouse embryos, *Znrf3* knockout caused open neural tube defects, which is a common consequence of the Wnt/PCP signaling disruption (*Hao et al., 2012*). Other report showed a similar phenotype in *Xenopus* embryos after *Rnf43* mRNA injection (*Tsukiyama et al., 2015*). And finally, in *Caenorhabditis elegans*, the homolog of RNF43 and ZNRF3 called plr-1 was shown to control not only the surface localization of frizzled, but also proteins related to mammalian noncanonical Wnt co-receptors ROR1/2 and RYK (*Moffat et al., 2014*). However, it is worth underlining that RSPO family homologs are absent in *C. elegans* (*Lebensohn and Rohatgi, 2018*), so the mode of action of RNF43/ZNRF3 in the worm might be different than in mammalian cells.

In this study, we have directly addressed the role of RNF43 in the WNT5A-induced signaling. We demonstrate that RNF43 controls the noncanonical Wnt pathway similarly to Wnt/β-catenin pathway. Further, we show that RNF43 is a relevant inhibitor of pro-metastatic WNT5A signaling in melanoma where it prevents both WNT5A-induced invasive behavior and WNT5A-assisted development of resistance to B-RAF and MEK inhibitors.

## Results

### RNF43 inhibits WNT5A-driven noncanonical Wnt signaling pathway

In order to test whether or not RNF43/ZNRF3 controls noncanonical Wnt signaling, we have decided to study T-REx 293 cells. T-REx 293 cells secrete endogenous WNT5A that constitutively activates the noncanonical Wnt pathway – this can be demonstrated by the CRISPR/Cas9-mediated knockout of *WNT5A* (*Kaiser et al., 2020*). Removal of endogenous WNT5A in T-REx 293 cells is sufficient to eliminate the activation of readouts of WNT5A signaling such as phosphorylation of ROR1, DVL2, and DVL3 that can be monitored by western blotting as the decrease in the phosphorylation-mediated electrophoretic mobility shifts (*Figure 1A*; *Bryja et al., 2007b*; *Kotrbová et al., 2020*; *Radasz-kiewicz and Bryja, 2020*). Autocrine WNT5A signaling was promoted by the inhibition of endogenous RNF43/ZNRF3 by RSPO1 treatment (*Figure 1B*, compare lanes 1 and 2) and inhibited by RNF43 overexpression under the tetracycline (Tet)-controlled promoter (TetON) or by block of WNT secretion using Porcupine inhibitor Wnt-C59 (*Figure 1B*). To confirm that the effects are indeed caused by the block of WNT5A signaling, T-REx 293 cells pretreated with Wnt-C59 and as such unable to produce Wnt ligands were stimulated with increasing doses of recombinant WNT5A. As shown in *Figure 1C*, overexpression of RNF43 completely blocked signaling induced by recombinant WNT5A. Altogether, this demonstrates that RNF43 has the potential to block WNT5A signaling in mammalian cells.

### RNF43 physically interacts with key proteins from the noncanonical Wnt pathway

To address the molecular mechanism of RNF43 action in the noncanonical Wnt pathway, we decided to describe RNF43 interactome by the proximity-dependent biotin identification (BioID; *Roux et al., 2012*), which was already successfully applied in the challenging identification of E3s substrates (*Coyaud et al., 2015*; *Deshar et al., 2016*). We have exploited our recently published dataset (*Spit et al., 2020*) based on T-REx 293 TetON cells that inducibly expressed RNF43 fused C-terminally (intracellularly) with BirA* biotin ligase. Several core proteins of the noncanonical Wnt signaling pathway – namely, ROR1, ROR2, VANGL1, VANGL2, SEC24B, and all three isoforms of DVL – were strongly and specifically biotinylated by RNF43-BirA* (*Figure 1D and D'*, *Figure 1—source data 1*). Furthermore, the noncanonical Wnt pathway was significantly enriched also in the gene ontology (GO) terms (*Figure 1—source data 2*). Altogether, it suggests that RNF43 can at least transiently interact with multiple proteins involved in the Wnt/PCP pathway, including essential receptor complex components from the ROR, DVL, and VANGL protein families.

To validate the protein-protein interactions identified by BioID, we performed a series of co-immunoprecipitation (co-IP) and co-localization experiments (*Figure 2*, *Figure 2—figure supplement 1*). We have focused on the interactions of RNF43 with ROR1/ROR2 and with VANGL1/VANGL2 mainly because these interactions are novel and at the same time highly relevant for the noncanonical Wnt pathway. RNF43 co-immunoprecipitated with both VANGL2 (*Figure 2A*) and VANGL1 (*Figure 2—figure supplement 1A*). More detailed analysis of VANGL2 showed co-localization of VANGL2 and RNF43 in the cell membrane (*Figure 2B and B'*). RNF43 also efficiently pulled down ROR1 (*Figure 2C*) and ROR2 (*Figure 2—figure supplement 1B*). Deletion of the cysteine-rich domain (CRD) (ROR2, *Figure 2—figure supplement 1B*) had no impact on the amount of co-immunoprecipitated RNF43, which suggests that RNF43 primarily interacts with RORs intracellularly. Both ROR1/ROR2 co-localized with RNF43 at the level of the plasma membrane (*Figure 2D and D'* and *Figure 2—figure supplement 1C and C'*). It was described that RORs and VANGLs also bind DVL (*Gao et al., 2011*; *Mentink et al., 2018*; *Seo et al., 2017*; *Witte et al., 2010*; *Yang et al., 2017*) and at the same time DVL proteins mediate ubiquitination of FZD receptors by RNF43 in the Wnt/β-catenin pathway (*Jiang et al., 2015*). To address whether DVL also acts as a physical link between RNF43 and the analyzed PCP proteins, we performed the co-IP experiments with VANGL2 and ROR1 in the T-REx 293 cells lacking all free DVL isoforms (DVL triple knockout cells) (*Paclíková et al., 2017*). As shown in *Figure 2—figure supplement 1D and E*, RNF43 was able to bind both VANGL2 and ROR1 as efficiently as in the wild-type cells (compare with *Figure 2A and D*). In summary, our results indicate that RNF43 interacts, in a DVL-independent way, with PCP proteins from VANGL and ROR families.

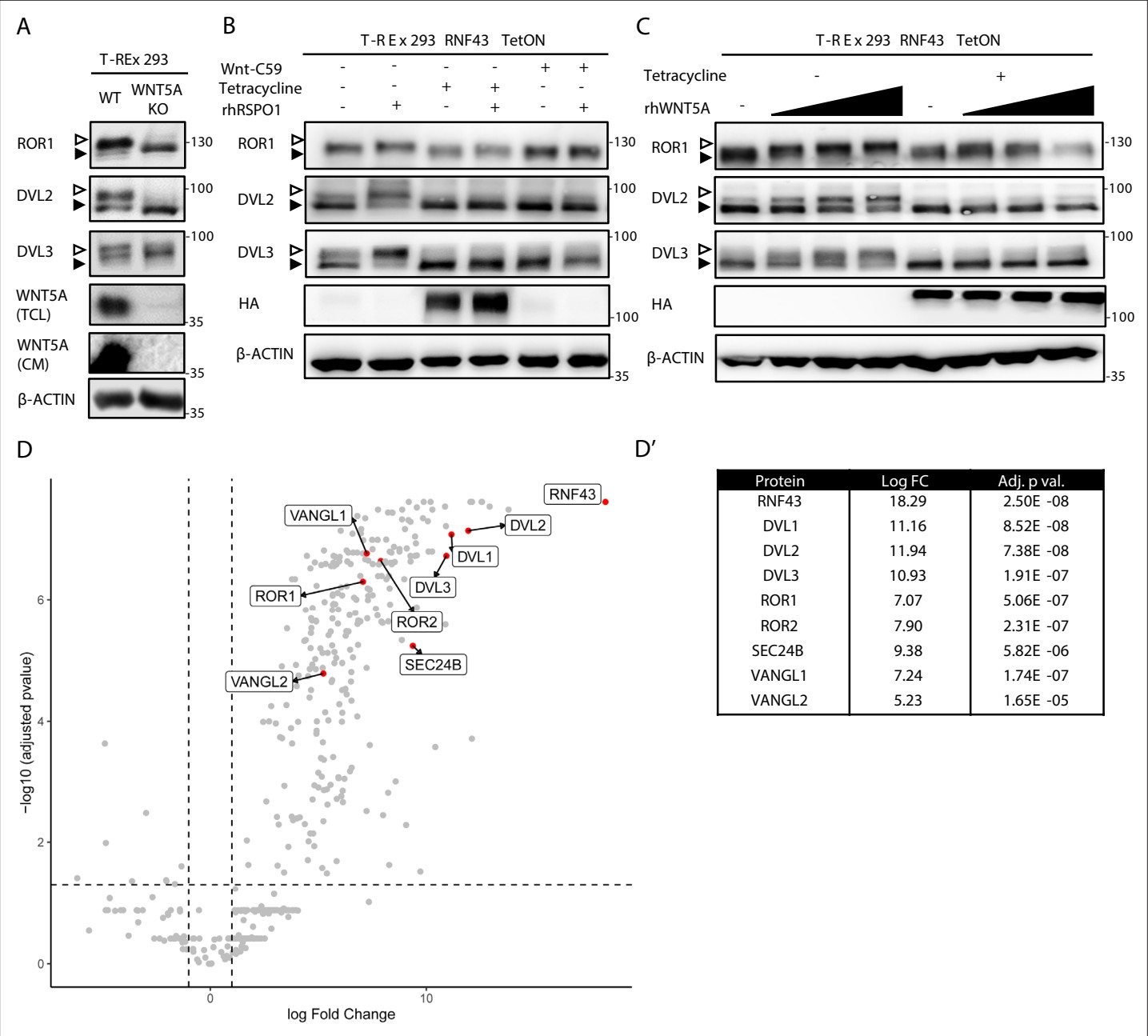

**Figure 1.** RNF43 interactome is enriched with the Wnt planar cell polarity pathway components. (**A**) Western blot analysis of T-REx 293 wild type (WT) and *WNT5A* KO cells. Phosphorylation-dependent shifts of endogenous ROR1, DVL2, and DVL3 were suppressed upon WNT5A loss (TCL: total cell lysate; CM: conditioned medium). Signal of β-actin serves as a loading control. Empty arrowhead marks phosphorylation-dependent shift; black arrowhead indicates unphosphorylated protein. (**B**) Western blot showing activation of the noncanonical Wnt pathway components: ROR1, DVL2, and DVL3 upon rhRSPO1 overnight treatment (arrowheads as in **A**). Tetracycline-forced RNF43 overexpression (as visualized by HA tag-specific antibody) suppressed this effect. Inhibition of Wnt ligands secretion by the Porcupine inhibitor Wnt-C59 shows dependency of the rhRSPO1-mediated effect on endogenous Wnt ligands; representative blots from N = 3. (**C**) Western blot analysis of cellular responses to the increasing doses of rhWNT5A. ROR1 shift and phosphorylation of DVL2 and DVL3 (empty arrowhead) were inhibited upon tetracycline-induced RNF43-HA-BirA* overexpression. All samples were treated with Wnt-C59 Porcupine inhibitor to ascertain assay specificity to the exogenous rhWNT5A, N = 3. (**D**) Volcano plot of the RNF43 interactome identified by BioID and subsequent mass spectrometric detection (see Materials and methods for details). Significantly enriched proteins annotated as the components of the noncanonical Wnt signaling pathway are highlighted and their log fold change and adjusted p values are presented (**D'**). A full list of BioID-based identified interactors of RNF43 is presented in *Figure 1—source data 1* and GO terms enrichment analysis in *Figure 1—source data 2*.

The online version of this article includes the following source data for figure 1:

*Figure 1 continued on next page*

*Figure 1 continued*

**Source data 1.** BioID RNF43 interactors.

**Source data 2.** gProfiler GO terms analysis.

## RNF43 ubiquitinates VANGL2 and triggers its degradation

Since RNF43 is an E3 ubiquitin ligase, we next tested whether it can ubiquitinate its binding partners from the noncanonical Wnt pathway. Enzymatically inactive RNF43 Mut1 variant (*Koo et al., 2012*) served here as a negative control. Using His-ubiquitin pulldown assay, we were able to show that VANGL2 (*Figure 3A*), as well as DVL1 and DVL2 (*Figure 3—figure supplement 1A*), was ubiquitinated when co-expressed with RNF43 but not with RNF43 Mut1. However, we were unable to detect RNF43-induced ubiquitination of ROR1 or ROR2 (negative data, not shown).

Further analysis showed that overexpression of RNF43, but not its E3 ligase dead variant, decreased VANGL2 protein level (*Figure 3B*, quantified in *Figure 3—figure supplement 1B*). Decrease in VANGL2 caused by RNF43 was accompanied by impeded phosphorylation of ROR1 (*Figure 3B*) and DVL3 (*Figure 3B*, *Figure 3—figure supplement 1B*). On the other side, two independent clones of cells deficient in both RNF43 and ZNRF3 (*RNF43/ZNRF3* dKO; *R/Z* dKO) showed higher VANGL2 levels and higher DVL phosphorylation (*Figure 3B*, *Figure 3—figure supplement 1B*). These confirm that also endogenous RNF43/ZNRF3 module affects the noncanonical Wnt signaling. Interestingly, treatment with the proteasome inhibitor MG132 but not with autophagosome-lysosome inhibitor chloroquine blocked these effects of RNF43 (*Figure 3C*). This suggests that RNF43 action in the noncanonical Wnt pathway depends on the proteasomal degradation pathway, which differs from the Wnt/β-catenin pathway, where RNF43 triggers FZD degradation via the lysosomal pathway (*Koo et al., 2012*). This is in accordance with immunofluorescence analysis, which did not show an increased co-localization of VANGL2 and RNF43 in lysosomes (*Figure 3—figure supplement 2B*). However, RNF43 overexpression led to the retention of VANGL2 in the Golgin-97-positive area (*Figure 3—figure supplement 2C*). This suggests that RNF43 can act via similar mechanism that was described for WNT5A-induced regulation of VANGL2 plasma membrane localization during planar cell polarity maintenance (*Feng et al., 2021*; *Guo et al., 2013*; *Tower-Gilchrist et al., 2019*; *Yang et al., 2017*).

## RNF43 induces ROR1 endocytosis by a clathrin-dependent pathway

ROR1 and ROR2 are the key receptors for WNT5A that we found to interact with RNF43 (*Figures 1 and 2*). We thus speculated that RNF43 can regulate ROR1/ROR2 surface levels. T-REx cells express dominantly ROR1, and indeed flow cytometric analysis demonstrated that cell lacking endogenous RNF43 and ZNRF3 have more ROR1 receptor on the surface than parental T-REx cells (*Figure 3D*). The staining is specific as demonstrated by the validation of the ROR1-APC antibody in *ROR1* KO T-REx 293 cells (*Figure 3—figure supplement 1C and D*). When we introduced inducible RNF43 into *RNF43/ZNRF3* dKO T-REx cell line, we were able to rescue this phenotype, and after 3 hr of tetracycline treatment, we detected decreased surface ROR1 and the overnight exposition to tetracycline had no significant effect (*Figure 3E and E'*).

In our analysis of RNF43 interactors (*Figure 1D*), we identified also multiple proteins involved in endosomal transport. It included proteins involved in the clathrin endocytic pathway – STAM1, HRS, ZFYVE16, PICALM, NUMB, RAB11-FIP2, and subunits of the associated adaptor protein complexes AP-3 and AP-4 (*Supplementary file 1*; *Bache et al., 2003*; *Cullis et al., 2002*; *Hirst et al., 2013*; *Raiborg et al., 2001*; *Santolini et al., 2000*; *Seet and Hong, 2005*; *Tebar et al., 1999*). Based on the BioID results analysis, we speculated that RNF43 may promote clathrin-mediated endocytosis of ROR1. Thus, we applied dansylcadaverine to block this pathway (*Blitzer and Nusse, 2006*). In agreement with our hypothesis, treatment with this inhibitor prevented RNF43-mediated effect on the ROR1 surface expression in T-REx 293 RNF43/ZNRF3 dKO RNF43 TetON cells (*Figure 3F*).

To get a better insight into the mechanism of RNF43-induced internalization of ROR1, we analyzed the co-localization of ROR1 and RAB5 (marker of early endosomes) and RAB11 (marker of recycling endosomes) in T-REx 293 R/Z dKO RNF43 TetON (*Figure 3G* and *Figure 3—figure supplement 3A*) and T-REx 293 RNF43 TetON cells (*Figure 3—figure supplement 1E* and *Figure 3—figure supplement 3B*). Hyperactivation of Rab5 by overexpression of wild-type Rab5 leads to the formation of giant early endosomes (*Bucci et al., 1992*) where we observed ROR1/RAB5 co-localization after 3

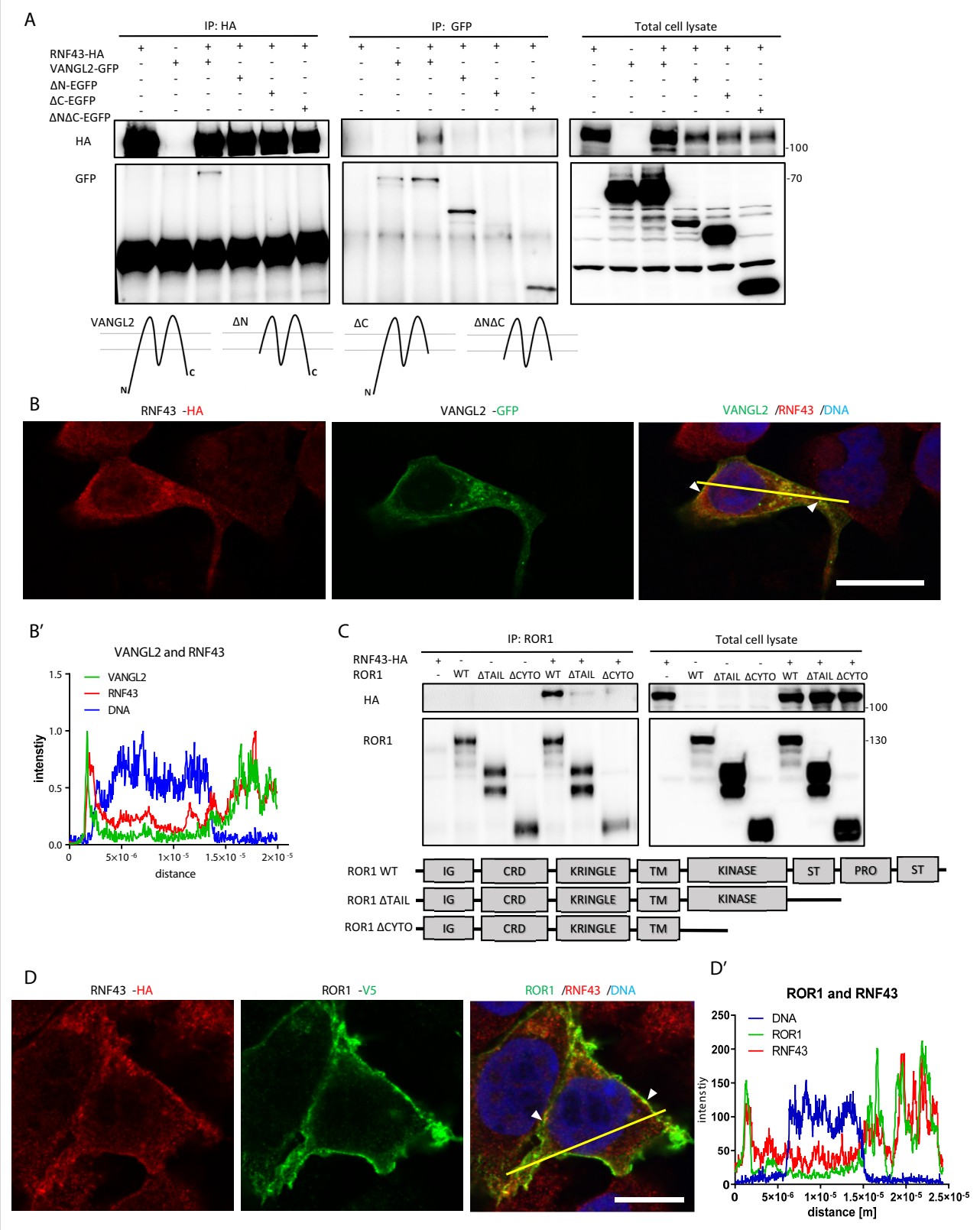

**Figure 2.** RNF43 interacts with Wnt/planar cell polarity (PCP) components. (**A**) RNF43 interacts with VANGL2, but not with its mutants lacking N- or C-termini. VANGL2-EGFP and its variants (schematized) were overexpressed with RNF43-HA in Hek293 T-REx cells, immunoprecipitated by anti-HA and anti-GFP antibodies and analyzed by western blotting. Representative experiment from N = 3. Scheme illustrates secondary structure of the wild-type VANGL2 protein and its shortened variants used in this study. (**B**, **B'**) RNF43 (anti-HA, red) co-localized with transiently expressed VANGL2 (GFP, green).

*Figure 2 continued on next page*

*Figure 2 continued*

Co-localization was analyzed utilizing histograms of red, green, and blue channels signals along selection (yellow line) (**B′**). TO-PRO-3 Iodide was used to stain nuclei (blue). Scale bar: 25 µm. (**C**) RNF43 binds to the ROR1 and deletion of the intracellular part of ROR1 disrupts this interaction. RNF43-HA was detected in the ROR1 pull down prepared from lysates of Hek293 T-REx cells overexpressing RNF43-HA and ROR1-V5, N = 3. ROR1 wild-type and truncated mutants are represented in the scheme. (**D, D′**) RNF43 (anti-HA, red) co-localized with transiently expressed ROR1-V5 (anti-V5, green). Signals along selection (yellow line) were analyzed (**D′**). TO-PRO-3 was employed nuclei staining (blue). Scale bar: 25 µm. RNF43 interactions with VANGL1 and ROR2 are studied in *Figure 2—figure supplement 1*.

The online version of this article includes the following source data and figure supplement(s) for figure 2:

**Source data 1.** RNF43 interacts with Wnt/planar cell polarity (PCP) components.

**Figure supplement 1.** RNF43 interacts with Wnt/planar cell polarity (PCP) components.

**Figure supplement 1—source data 1.** RNF43 interacts with Wnt/planar cell polarity (PCP) components.

hr of tetracycline treatment. The co-localization decreased after overnight exposition to tetracycline. RAB11$^+$ endosomes were recruited to the ROR1 as well as after RNF43 induction and RAB11 co-localized strongly with ROR1 even after ON treatment. We conclude that surface ROR1 is controlled by RNF43 via interference with RAB5- and RAB11-mediated endocytosis and RNF43-mediated effect is not persistent due to the activity of recycling RAB11 recycling pathway (*Ullrich et al., 1996*).

## RNF43 expression is decreased in human melanoma

Our data shown in *Figures 1–3* demonstrate that RNF43 can inhibit WNT5A-induced noncanonical signaling via downregulation of the receptor complexes. But is RNF43 capable of blocking WNT5A-induced biological processes? WNT5A signaling plays a crucial role in melanoma, one of the most malignant tumor types. High expression of *WNT5A* in this cancer is a negative overall survival and positive metastasis formation factor (*Da Forno et al., 2008*; *Luo et al., 2020*; *Weeraratna et al., 2002*). Signaling cascade activated by WNT5A in melanoma drives epithelial–mesenchymal transition-like program (EMT), resulting in increased metastatic properties of melanoma cells in vitro and in vivo (*Dissanayake et al., 2008*; *Dissanayake et al., 2007*; *Sadeghi et al., 2018*). In melanoma, WNT5A acts through FZD and ROR1/ROR2 (*O'Connell et al., 2010*; *Tiwary and Xu, 2016*; *Weeraratna et al., 2002*). The importance of WNT5A-driven signaling in melanoma is thus well recognized, and melanoma represents probably the most characterized (and most clinically relevant) pathophysiological condition where noncanonical WNT5A signaling drives cell invasion and disease progression (*Arozarena and Wellbrock, 2017b*; *Da Forno et al., 2008*; *Dissanayake et al., 2007*; *Lai et al., 2012*; *Liu et al., 2018*; *O'Connell et al., 2010*; *O'Connell et al., 2008*; *Weeraratna et al., 2002*).

Interestingly, the in silico analysis of gene expression (*Talantov et al., 2005*; *Xu et al., 2008*) showed that *RNF43* expression dramatically decreases between benign melanocytic skin nevus and cutaneous melanoma (*Figure 4A*; *Talantov et al., 2005*) and further between primary site and metastasis (*Xu et al., 2008*; *Figure 4B*). Importantly, analysis of other datasets (*Anaya, 2016*) showed that *RNF43* low melanoma patients have shorter overall survival (OS; *Figure 4C*). *ZNRF3* expression had no prognostic value (*Figure 4—figure supplement 1A*). Interestingly, the expression of two genes encoding direct targets ubiquitinated by RNF43, namely, *DVL3* and *VANGL1*, increased during melanoma progression (*Figure 4—figure supplement 1B and C*) and high expression in both cases correlates with bad prognosis and shorter overall survival (*Figure 4D*, *Figure 4—figure supplement 1D*). All these findings are in line with the hypothesis that RNF43 acts in melanoma as a tumor suppressor that restricts WNT5A-induced biological processes and gets silenced during melanoma progression.

## RNF43 inhibits invasive properties of melanoma cells in vitro

A375 and A2058 are human melanoma cell lines carrying BRAF V600E mutations that are broadly used to study WNT5A role in melanoma (*Anastas et al., 2014*; *Connacher et al., 2017*; *Da Forno et al., 2008*; *Ekström et al., 2014*; *Linnskog et al., 2016*; *Liu et al., 2018*). For the purpose of our study, we chose A375 wild-type (A375) cells and their derivate with the increased metastatic potential referred to as A375 IV (*Kucerova et al., 2014*). Both A375 variants and A2058 express *WNT5A*, *RNF43*, and *ZNRF3* (*Figure 4—figure supplement 2A–C, I–K*). A375 and A375 IV secrete WNT5A and WNT11 to the culture medium, whereas A2058 produces only WNT11 (*Figure 4E*). WNT5A protein levels are higher and WNT11 levels lower in case of A375 IV metastatic derivate in comparison to parental A375

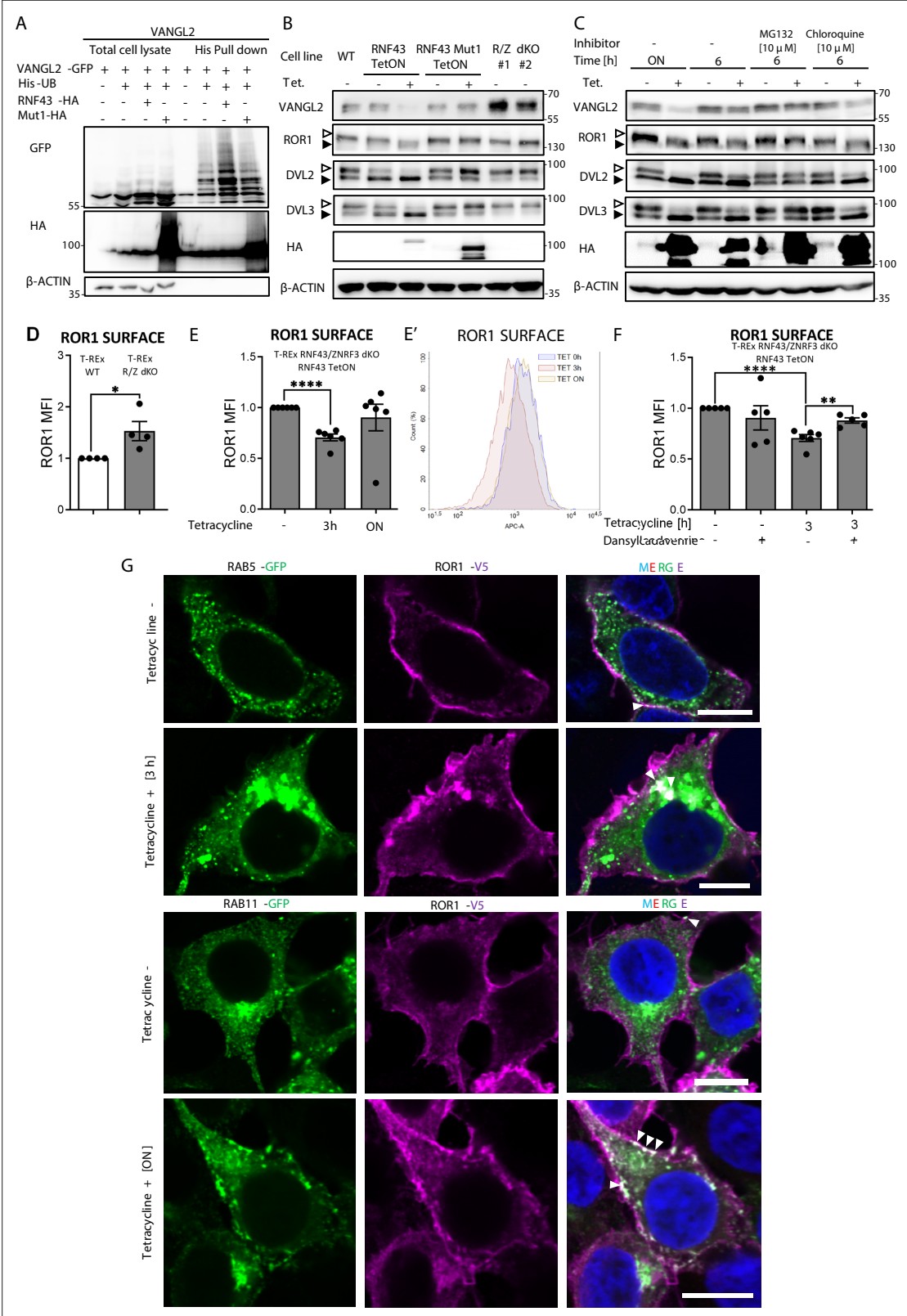

**Figure 3.** Mechanism of Wnt/planar cell polarity (PCP) inhibition by RNF43. (**A**) Hek293 T-REx cells were transfected with plasmid encoding His-tagged ubiquitin, VANGL2-GFP and HA-tagged wild-type or Mut1 RNF43 constructs. Ubiquitinated proteins were enriched by His pull down and analyzed by western blotting. VANGL2 is ubiquitinylated by the E3 ubiquitin ligase RNF43, but not by its enzymatically inactive variant (RNF43Mut1). Representative experiment from N = 3. RNF43-mediated ubiquitination of DVL1 and DVL2 together in *Figure 3—figure supplement 1*. (**B**) Tetracycline-induced

*Figure 3 continued on next page*

*Figure 3 continued*

overexpression of the wt RNF43 (HA), but not enzymatically inactive RNF43Mut1 (HA), decreased VANGL2 protein level and suppressed phosphorylation of ROR1 and DVL3 (empty arrowhead; full arrowhead indicates unphosphorylated protein). CRISPR/Cas9-derived *RNF43/ZNRF3* (R/Z) dKO cell lines #1 and #2 display phenotype reversed to the RNF43 overexpression. Quantified in *Figure 3—figure supplement 1B*, N = 3. (**C**) Inhibition of the proteasomal degradation pathway by MG132 (but not by lysosomal inhibitor chloroquine) blocked the RNF43 effects on ROR1, DVL2, DVL3 (empty arrowhead: phosphorylated; full arrowhead indicates unphosphorylated protein) and VANGL2 as shown by the western blotting analysis, N = 3. (**D**) Flow cytometric analysis of surface ROR1 in wild-type (WT) and *RNF43/ZNRF3* (R/Z) dKO cells; unpaired two-tailed t-test: p=0.0298, N = 4. ROR1 was stained using ROR1-APC conjugate on the nonpermebilized cells. Validation of the a-ROR1-APC antibody is shown in *Figure 3—figure supplement 1D*. (**E**, **E''**) Surface ROR1 levels upon 3 hr and overnight (ON) induction of RNF43 in RNF43 TetON *RNF43/ZNRF3* dKO cells; unpaired t-test p<0.0001, N = 6. Representative histogram of ROR1-APC signal in the analyzed conditions is shown (**E'**). (**F**) Dansylcadaverine, inhibitor of clathrin-mediated endocytosis, blocked the effect of RNF43 overexpression on surface ROR1, performed as in (**E**); unpaired t-test p=0.0037 3 hr Tet. vs 3 hr Tet.+ dansylcadaverine; N = 5 and 6 (3 hr Tet. condition). (**G**) Immunofluorescence imaging showed enhanced ROR1 (anti-V5, magenta) co-localization with the marker of early endosomes RAB5 (GFP, green) after 3 hr tetracycline treatment in RNF43 TetON *RNF43/ZNRF3* dKO cells. RAB11-positive (GFP, green) recycling endosomes were recruited to the ROR1 (anti-V5, magenta) at the plasma membrane after overnight tetracycline treatment. Cells were transfected, treated, fixed, and stained. DNA was visualized by Hoechst 33342 (blue). Other tetracycline time points are shown in *Figure 3—figure supplement 3A*. Similar results were obtained for T-REx RNF43 TetON cell line (*Figure 3—figure supplement 1E*). Raw data used in (**D–F**) are given in *Figure 3—source data 1*.

The online version of this article includes the following source data and figure supplement(s) for figure 3:

**Source data 1.** Mechanism of Wnt/planar cell polarity (PCP) inhibition by RNF43.

**Figure supplement 1.** Mechanism of Wnt/planar cell polarity (PCP) inhibition by RNF43.

**Figure supplement 1—source data 1.** Mechanism of Wnt/planar cell polarity (PCP) inhibition by RNF43.

**Figure supplement 2.** Mechanism of Wnt/planar cell polarity (PCP) inhibition by RNF43.

**Figure supplement 3.** Mechanism of Wnt/planar cell polarity (PCP) inhibition by RNF43.

cell line (*Figure 4E' and E''*). Interestingly, *RNF43* expression in the A375 IV cells was significantly lower than that in the A375 parental cells (*Figure 4—figure supplement 2B*). Expression of *ZNRF3* did not differ and it was not affected by RNF43 overexpression (*Figure 4—figure supplement 2C*). Further, A375 line has lower expression of *ROR1* than IV derivate (*Figure 4—figure supplement 2D*). A2058 cells are *ROR1* and *ROR2* positive with *ROR1* level being higher (*Figure 4—figure supplement 2L and M*).

To study the RNF43 function, we generated A375 cells lacking *RNF43/ZNRF3* (*R/Z* dKO) by CRISPR/Cas9 method (sequencing results are presented in *Supplementary file 1*), cells stably over-expressing RNF43 (RNF43 OE) as well as A375, A2058, and additionally NRAS^{Q61L/WT} HRAS^{G13D/G13D} MelJuso -based doxycycline-inducible RNF43 TetON lines (*Figure 4F*). The initial characterization of A375 derivatives essentially confirmed the findings from T-REx 293 (see *Figure 1*), where RNF43 loss- and gain of function correlated strongly with the level of Wnt pathway activation assessed as DVL phosphorylation (*Figure 4G*, quantified in *Figure 4—figure supplement 1E and F*). Total protein levels of DVL2, DVL3, as well as expression of their genes remained unaffected by the RNF43 manip-ulation (*Figure 4—figure supplement 2E–H*). Similarly to T-REx 293 cells, in A375 (*Figure 4H*), A375 IV (*Figure 4I*), and A2058 (*Figure 4J* and quantified effect of RNF43 OE on the endogenous WNT pathway activity in *Figure 4—figure supplement 1G and H*) melanoma cells RNF43 overexpression efficiently blocked WNT5A-induced signaling. To extend the importance of our studies further, we tested the RNF43-mediated effect also in the *RAS*-mutant MelJuso cells. Here we also confirmed the suppression of ROR1, DVL2, and DVL3 shifts by forced RNF43 expression (*Figure 4—figure supplement 3A*). Inducible model of A375 RNF43 TetON cells drew a similar picture (*Figure 4—figure supplement 3B*), compared to the control cells that underwent the same procedures (*Figure 4—figure supplement 3C*).

WNT5A signaling has been related to numerous biological features that support the invasive prop-erties of melanoma (*Arozarena and Wellbrock, 2017b*; *O'Connell and Weeraratna, 2009*; *Prasad et al., 2015*; *Weeraratna et al., 2002*). To address if RNF43 affects any of these WNT5A-controlled properties, we have compared parental and RNF43-derivative melanoma cells in a panel of functional assays that included (1) wound healing assay, (2) collagen I hydrogel 3D chemotaxis assay, (3) Matrigel invasion assay, (4) invadopodia formation assay, and (5) gelatin degradation assay. Firstly, all A375, A375 IV, and A2078 cells overexpressing RNF43 showed suppressed 2D collective migration in the wound healing assay (*Figure 5A–E*). Next, we analyzed the impact of RNF43 expression on directional

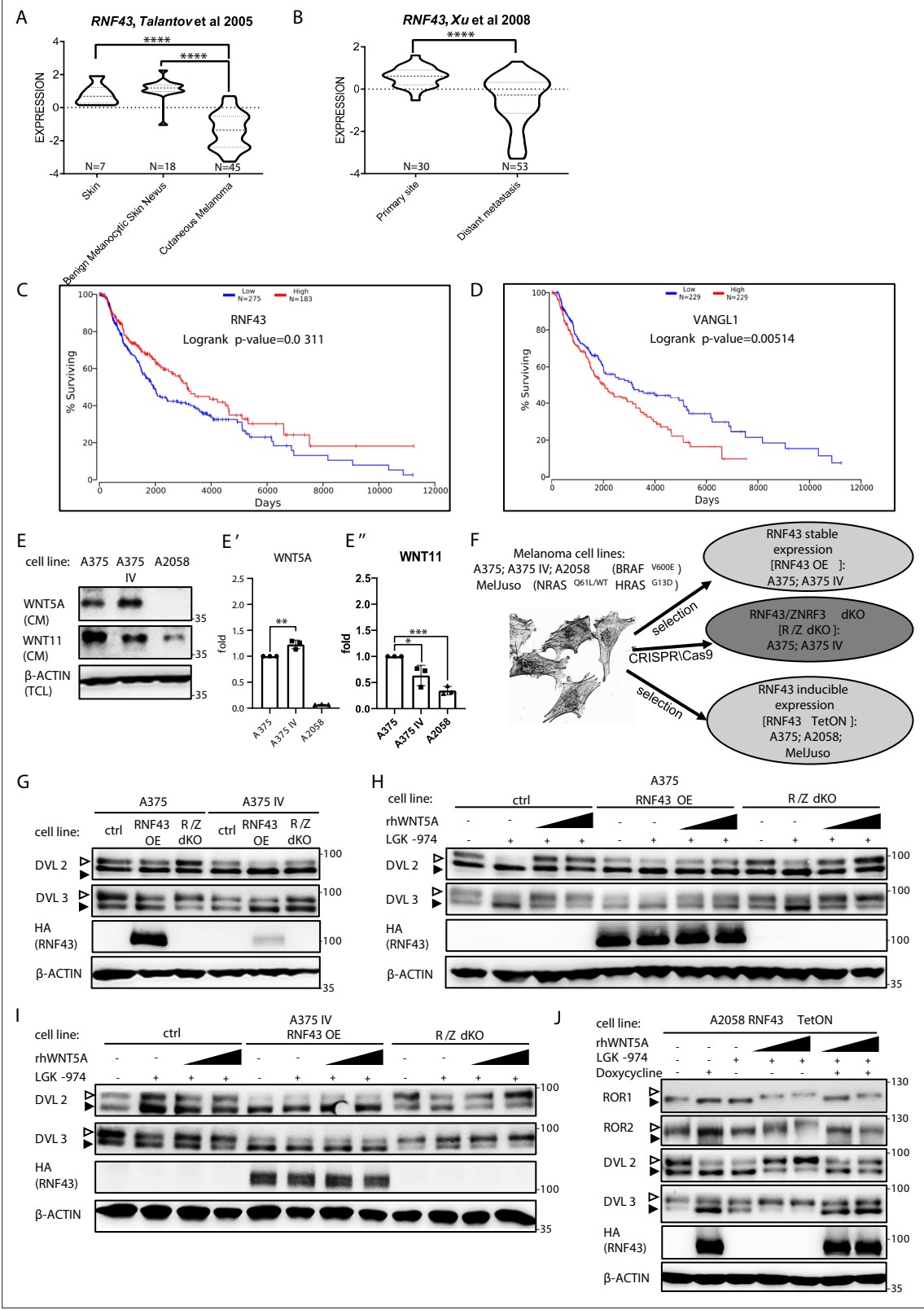

**Figure 4.** RNF43 in melanoma. (**A**, **B**) *RNF43* expression is lower in melanoma when compared with the skin and benign melanocytic skin nevus (**A**) and in the case of distant metastasis compared to the primary tumors (**B**), unpaired two-tailed t-test: ****p<0.0001. (**C**, **D**) *RNF43* expression is a negative prognostic factor in melanoma. *RNF43* low patients have shorter overall survival (logrank p-value=0.0311). Contrary, patients with low-expression *VANGL1* (**D**) had longer survival (logrank test, p-value=0.00518). Expression of *DVL3*, *VANGL1*, and *ZNRF3* is analyzed in ***Figure 4—figure***

*Figure 4 continued on next page*

Figure 4 continued

*supplement 1A–D*. (**E**, **E'**, **E"**) Culture media from melanoma A375, A375 IV, and A2058 cell lines were collected after 48 hr and analyzed by western blotting for the presence of WNT5A (**E'**) and WNT11 (**E"**). Densitometric analysis has been done using the ImageJ software. Graphs represent ratio of corresponding WNT (medium) and β-actin (lysate) signals. Unpaired two-tailed t-test: p=0.0092 (**E'**); 0.0320 and 0.0001 (**E"**); N = 3. (**F**) Schematic representation of the melanoma cell lines and their genetically modified variants used in this study. (**G**) Effects of the stable RNF43 overexpression and *RNF43/ZNRF3* knockout in A375 and in its invasive derivate A375 IV. Exogenous RNF43 expression blocked DVL2 and DVL3 activation (arrowheads showing phosphorylation-dependent change of the electrophoretic mobility). Removal of endogenous RNF43 and ZNRF3 proteins had an opposite effect, N = 6. Quantification is given in *Figure 4—figure supplement 1E,F*. Expression of *WNT5A, RNF43, ZNRF3, ROR1, DVL2,* and DVL3 in tested cell lines was checked and shown in *Figure 4—figure supplement 2*. (**H–J**) Western blot showing inhibitory effect of RNF43 on A375 (**H**), A375 IV (**I**), and A2058 RNF43 TetON (**J**) cell lines in response to the 40 and 80 ng/ml 3 hr-long rhWNT5A treatments. *RNF43/ZNRF3* dKO A375 (**H**), A375 IV (**I**) cell lines had stronger response to rhWNT5A than parental line. β-actin served as a loading control. Empty arrowhead: phosphorylated; full arrowhead: unphosphorylated protein. Porcupine inhibitor LGK-974 was used to block endogenous Wnt ligands secretion and RNF43 was probed by HA antibody, N = 3 and N = 4 (A2058). Quantification of DVL2 and DVL3 activation status and their protein levels in A2058 cells is given in *Figure 4—figure supplement 1G and H*. *Figure 4—figure supplement 3* presents rhWNT5A treatment and consequences of RNF43-induced expression in RAS-mutant melanoma cell line MelJuso (**A**) and A375 (**B**) together with isogenic control (**C**). *Figure 4—source data 1* contains raw data.

The online version of this article includes the following source data and figure supplement(s) for figure 4:

**Source data 1.** RNF43 in melanoma.

**Figure supplement 1.** RNF43 in melanoma.

**Figure supplement 1—source data 1.** RNF43 in melanoma.

**Figure supplement 2.** RNF43 in melanoma.

**Figure supplement 2—source data 1.** RNF43 in melanoma.

**Figure supplement 3.** RNF43 in melanoma.

**Figure supplement 3—source data 1.** RNF43 in melanoma.

invasion through the collagen I hydrogel in response to chemokines. This assay mimics the taxis of melanoma cells in body: CCL21 drives lymph nodes metastasis and CXCL12 promotes lung invasion (*Figure 5F*; *Jacquelot et al., 2018*; *McArdle et al., 2016*). A375 IV and A2058 cell lines showed significant response to these treatments and RNF43 blocked completely these invasion events (*Figure 5G and H*). Importantly, the response of A375 cells in this assay was negligible (*Figure 5—figure supplement 1B*), confirming their decreased metastatic capacity compared to the A375 IV. In line, invasion of individual cells through the extracellular matrix (ECM) mimicking Matrigel was higher in A375 IV in comparison to A375 but in both cases reduced by RNF43 OE (*Figure 5—figure supplement 1D*). The same is true for the number of invadopodia-specialized structures mediating adhesion and remodeling of the surrounding ECM (*Eddy et al., 2017*; *Masi et al., 2020*). A375 IV cells formed more invadopodia than A375 parental cells (*Figure 5I*) and cells overexpressing RNF43 formed less of them (*Figure 5I*). In agreement, we also observed reduced gelatin degradation activity in A375 and A375 IV cells overexpressing RNF43 (*Figure 5J*). Furthermore, treatment with WNT5A enhanced the gelatin degradation capacity of A375 cells, but not their RNF43-overexpressing derivate (*Figure 5J*). Representative images from the conducted assays are shown in *Figure 5—figure supplement 1*, *Figure 5—figure supplement 2*, *Figure 5—figure supplement 3*, *Figure 5—figure supplement 4*. All these assays strongly support the conclusion that RNF43 acts as a strong molecular inhibitor of WNT5A-triggered proinvasive features of melanoma. Interestingly, *RNF43/ZNFR3* removal did not lead to significant potentiation in these functional assays, which suggests that the noncanonical Wnt pathway-controlled pro-metastatic features of these cells are already close to its maximum. In agreement, treatment with WNT5A showed the effect only in case A375 cells in the gelatin degradation assay (*Figure 5J*).

## RNF43 prevents acquisition of resistance to BRAF V600E targeted therapy

The mitogen-activated protein kinase (MAPK) pathway is hyperactivated in melanoma (*Davies et al., 2002*) as a result of UV-induced mutations triggering constitutive activation of this signaling axis. The most common genetic aberration – *BRAF V600E* is a target of anti-melanoma therapy (*Akbani et al., 2015*; *Birkeland et al., 2018*; *Chapman et al., 2011*; *Flaherty et al., 2010*; *Hodis et al., 2012*; *Shain et al., 2015*). Drugs targeting mutated BRAF (e.g., vemurafenib/PLX4032) in melanoma

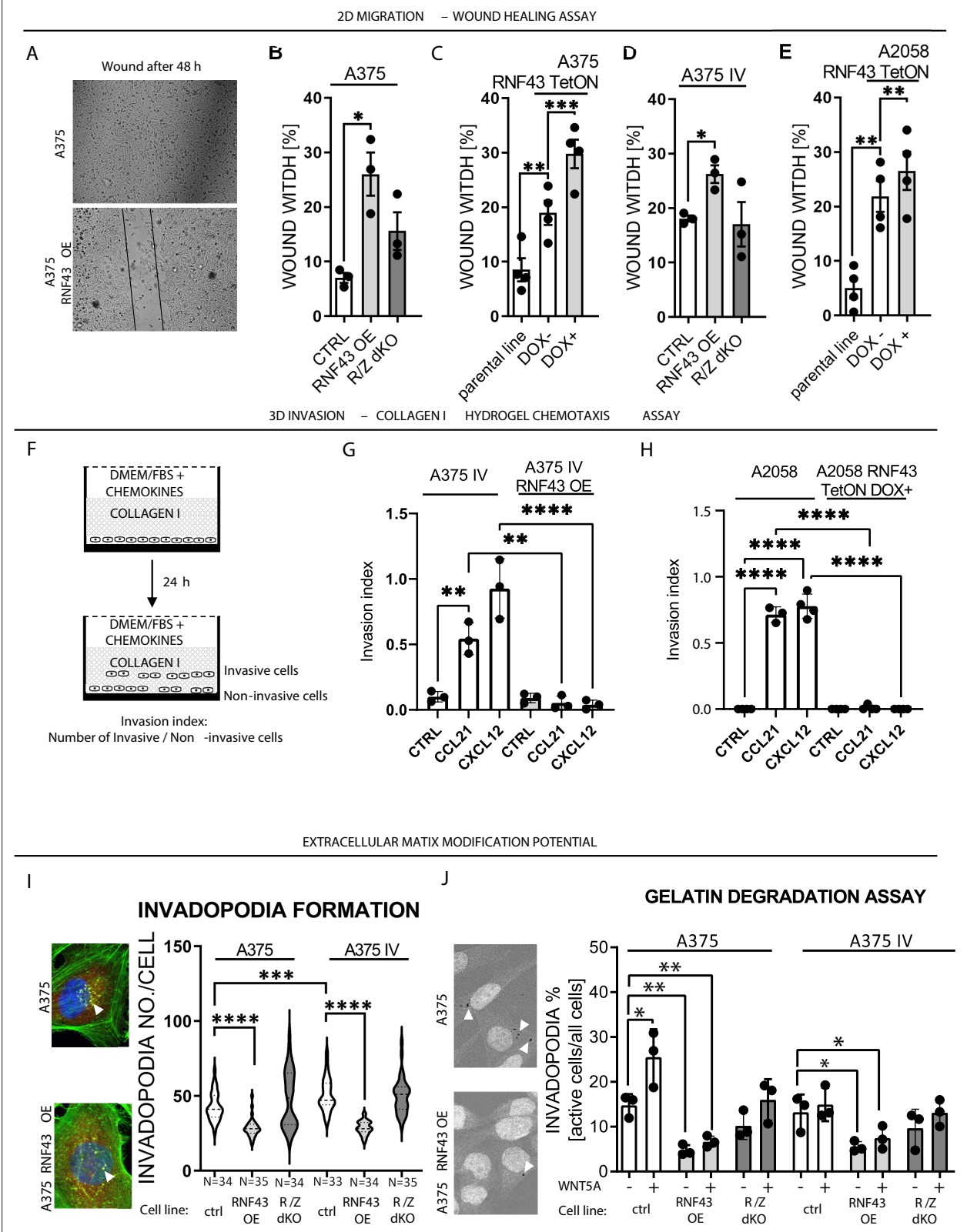

**Figure 5.** RNF43 inhibits WNT5A-dependent invasive properties of human melanoma. (**A–E**) RNF43 reduced migration of A375 (**B**), A375 RNF43 TetON (**C**), A375 IV (**D**), and A2058 RNF43 TetON (**E**) in the wound healing assay. Wound was photographed 48 hr after scratch and presented as % of cell-free surface at the end of the experiment. Cells proliferation was suppressed by serum starvation, unpaired two-tailed t-test: *p<0.05, **p<0.01, ***p<0.001, N = 3 (**B, D**) or 4 (**C, E**). Representative photos at the end of the experiment are shown in (**A**) and in *Figure 5—figure supplement 1A*. (**F–H**) RNF43

*Figure 5 continued on next page*

*Figure 5 continued*

blocked collagen I hydrogel 3D invasion in response to CCL21 (100 ng/ml) and CXCL12 (100 ng/ml) of A375 IV (**G**) and A2058 (**H**) cell lines. Cells were serum starved, collagen I (1.5 mg/ml) was overlaid and polymerized. Doxycycline was applied for RNF43 induction during starvation. After 24 hr, cells were fixed and stained for DNA (Hoechst 33342, blue) and F-actin (phalloidin, red) and imaged by confocal microscopy. Invasion index was calculated as the ratio of invaded cells at specified height to the number of noninvasive cells at the glass level, unpaired two-tailed t-test: **p<0.01, ****p<0.0001, N = 3 (**G**) or N = 4 (**H**). A375 cells did not invaded collagen I hydrogel (**Figure 5—figure supplement 1B**). Representative photos are presented in **Figure 5—figure supplement 1C and C'**. (**I**) RNF43 overexpression in A375 and A375 IV decreased number of invadopodia. Quantification of the invadopodia formed by melanoma cells, based on the analysis of confocal images. Number of cortactin/F-actin double-positive puncta in the individual cells was calculated in the ImageJ software, unpaired two-tailed t-test: ***p<0.001, ****p<0.0001. Examples of confocal imaging are shown: green, phalloidin; red, cortactin; blue, DNA. See **Figure 5—figure supplement 2** for images from all experimental conditions. (**J**) Gelatin degradation assay; both A375 and A375 IV RNF43-overexpressing cell lines showed decreased capacity to locally degrade the extracellular matrix. rhWNT5A treatment induced gelatin degradation by A375 cells. Serum-starved cells were plated onto gelatin-Oregon Green-coated coverslips and incubated for 24 hr. Images obtained by Leica SP8 confocal microscope were analyzed for the presence of gelatin degradation by individual cells using ImageJ software, unpaired two-tailed t-test: *p<0.05, **p<0.01, N = 3. Example of gelatin degradation is shown; more pictures are presented in **Figure 5—figure supplement 3** and **Figure 5—figure supplement 4**. Numerical data are given in **Figure 5—source data 1** file.

The online version of this article includes the following source data and figure supplement(s) for figure 5:

**Source data 1.** RNF43 inhibits WNT5A-dependent invasive properties of human melanoma.

**Figure supplement 1.** RNF43 inhibits Wnt5a-dependent invasive properties of human melanoma.

**Figure supplement 1—source data 1.** RNF43 inhibits Wnt5a-dependent invasive properties of human melanoma.

**Figure supplement 2.** RNF43 inhibits Wnt5a-dependent invasive properties of human melanoma.

**Figure supplement 3.** RNF43 inhibits Wnt5a-dependent invasive properties of human melanoma.

**Figure supplement 4.** RNF43 inhibits Wnt5a-dependent invasive properties of human melanoma.

have improved patients' survival (**Chapman et al., 2011**; **Flaherty et al., 2010**; **Joseph et al., 2010**). Unfortunately, patients receiving BRAF inhibitors (BRAFi) relapse after several months of monotherapy because of the acquired resistance (**Nazarian et al., 2010**). WNT5A was shown to play a crucial role in the process leading to the vemurafenib resistance (**Anastas et al., 2014**; **Arozarena and Wellbrock, 2019**; **Mohapatra et al., 2019**; **O'Connell et al., 2013**; **Prasad et al., 2015**; **Webster et al., 2015**). Therefore, we were interested in checking whether RNF43 inhibits, via its effects on WNT5A signaling, cellular plasticity in response to vemurafenib (PLX4032), a clinically used *BRAF* V600E inhibitor.

The process of vemurafenib resistance acquisition can be modeled in vitro. We applied an experimental scheme optimized for A375 (**Anastas et al., 2014**). This model (**Figure 6A**) allows to study both acute responses to vemurafenib (24 hr treatment) as well as the gradual adaptation of long-term cell culture to increasing vemurafenib doses. Vemurafenib-resistant (VR) cells can be obtained after approximately 2 months. As shown in **Figure 6B**, treatment with vemurafenib resulted in rapid and complete inhibition of ERK1/2 phosphorylation, the readout of MAPK activation (compare lanes 1 and 2). In contrast, A375 VR cells showed constitutive ERK1/2 phosphorylation in the presence of 2 µM vemurafenib (compare lanes 2 and 3). Interestingly, transient exposition to vemurafenib resulted in the impeded phosphorylation of ROR1, DVL2, and DVL3 and increased ROR2 protein level without change in its mRNA (**Figure 6B–D**, **Figure 6—figure supplement 1A and B**). On the other side, VR cells displayed elevated ROR1 levels (both protein and mRNA) and increased phosphorylation of DVL2 and DVL3 (**Figure 6B–D**, **Figure 6—figure supplement 1A and B**). Strikingly, the expression of *WNT5A* was also higher in the resistant VR cells (**Figure 6E**). This suggests that activation of noncanonical WNT5A-induced signaling and ROR1 and ROR2 changes is indeed a part of the melanoma adaptation mechanism to vemurafenib. Therefore, we challenged A375 and its RNF43-expressing derivatives with vemurafenib. As shown in **Figure 6F**, exogenous RNF43 decreased colony formation and proliferation of cells seeded in low density and vemurafenib further enhanced this effect. Importantly, both A375 and A375 IV-overexpressing RNF43 completely failed to develop resistance to vemurafenib and died off during selection at 1 µM vemurafenib concentration (**Figure 6G**). Both A375 and A375 IV *RNF43/ZNRF3* double KO cells survived selection with BRAFi but showed decreased proliferation rate. Lower proliferation of *RNF43/ZNRF3* dKO cells can be caused by senescence because these cells have elevated WNT5A pathway activity (i.e., **Figures 3B and 4I**, **Figure 4—figure supplement 1E and F**) and WNT5A at the same time causes a senescence-like phenotype of melanoma after exposition to vemurafenib (**Webster et al., 2015**). Alternatively, *RNF43/ZNRF3* KO could affect also canonical Wnt/β-catenin pathway that was shown to drive distinct features of melanoma

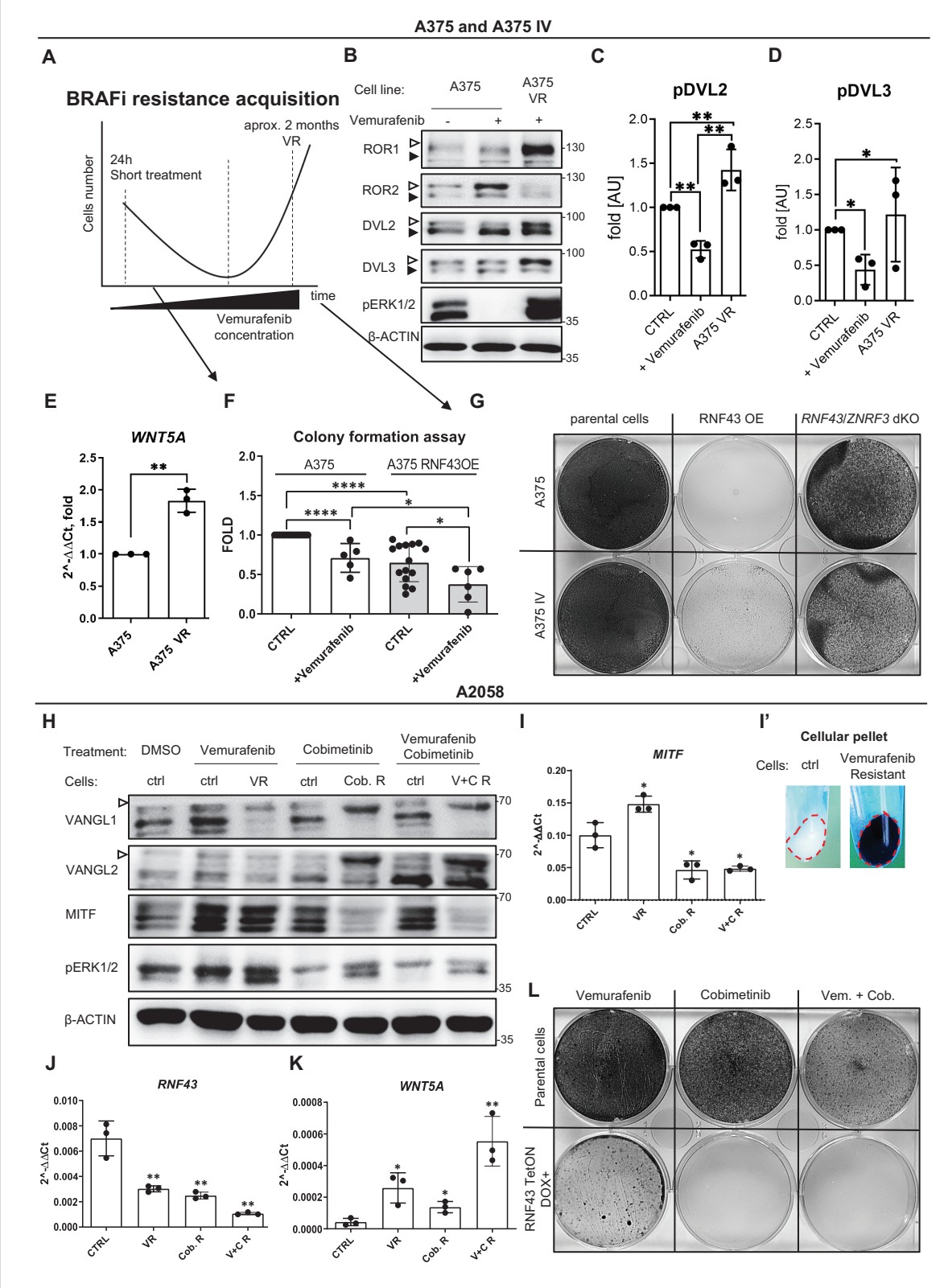

**Figure 6.** RNF43-overexpressing melanoma cells do not develop resistance to BRAF V600E targeted therapies. (**A**) Scheme showing the experimental model used for the analysis of vemurafenib resistance (VR) acquisition. Melanoma cells are exposed to the increasing doses of the *BRAF* V600E inhibitor vemurafenib and following initial decrease in cell numbers recover and obtain capacity to grow in the presence of vemurafenib. (**B–D**) Western blot analysis of the cellular responses to the acute vemurafenib treatment (0.5 µM, 24 hr) in comparison to the signaling in VR cells growing

*Figure 6 continued on next page*

*Figure 6 continued*

in the presence of 2 μM vemurafenib. Transient treatment resulted in decreased DVL2 and DVL3 phosphorylation and increased ROR2 signal. In VR cells, ERK1/2 is constitutively phosphorylated in the vemurafenib presence. β-actin served as a loading control. A375 VR cells showed increased activation of DVL2 and DVL3 (empty arrowheads; quantifications in **C** and **D**) and higher expression of ROR1. Unpaired two-tailed t-test: *p<0.05, **p<0.01, N = 3. *Figure 6—figure supplement 1A and B* presents changes in *ROR1* and *ROR2* genes expression and quantification of ROR1 and ROR2 proteins signals. (**E**) Expression of *WNT5A* gene is elevated in the VR A375 cells in comparison to the parental line, unpaired two-tailed t-test: **p=0.0013, N = 3. Data presented are normalized to 1. Relative expression was normalized to the *HSPCB and RPS13 genes* ($2^{-\Delta\Delta Ct}$). (**F**) Melanoma cell lines A375 and A375 IV overexpressing RNF43 showed decreased ability to grow and form colonies when seeded in the low density. After 7 days, colonies were fixed and stained with crystal violet. Paired (vemurafenib – vs+) and unpaired (A375 vs. IV) two-tailed t-tests: *p<0.05, ****p<0.0001, N ≥ 5. (**G**) RNF43-overexpressing A375 and A375 IV did not develop resistance to the BRAF V600E inhibition by vemurafenib treatment. Cells were cultured for approximately 2 months in the presence of increasing doses of the inhibitor. Photos show crystal violet-stained cultures at the end of the selection process. (**H**) A2058 cells (marked as 'ctrl') were exposed to the 24 hr-long treatments with vemurafenib (2.5 μM), cobimetinib (0.1 μM), or their combination. These short treatments were compared with cells cultured for approximately 2 months in the presence of vemurafenib (5 μM; VR), cobimetinib (0.5 μM; Cob. R) or their combination (V + C R). Experiment is schematized in *Figure 6—figure supplement 1C'*. Selected proteins were analyzed by western blot. Cells chronically exposed to inhibitors showed shifts of VANGL1 and VANGL2 bands (empty arrowhead) and changes in the MITF signal. Signal of pERK1/2 was used as treatments control. β-actin signal served as a loading control, N = 3. *Figure 6—figure supplement 1C* presents DVL2, DVL3 and total ERK1/2 signals. (**I–K**) Expression analysis of *MITF* (**I**), *RNF43* (**J**), and *WNT5A* (**K**) genes. *MITF* is significantly upregulated in cells resistant to 5 μM Vemurafenib and downregulated in cells derived from the long-term culture in the presence of 0.5 μM cobimetinib alone and in combination with vemurafenib. (**I'**) VR A2058 show higher pigmentation. *RNF43* expression decreased in the derived cells, while *WNT5A* increased. Relative expression levels are presented as $2^{-\Delta\Delta Ct}$ ± SD and normalized to the levels of *B2M* and *GAPDH* genes, unpaired two-tailed t-tests: *p<0.05, **p<0.01, N = 3. *Figure 6—figure supplement 1D–F* shows expression analysis of *ZNRF3, ROR1*, and *ROR2*. (**L**) RNF43-overexpressing A2058 cells did not develop resistance to the cobimetinib and its combination with vemurafenib. Cells were cultured for approximately 2 months in the presence of increasing doses of the inhibitors (up to 5 μM for vemurafenib and 0.5 μM for cobimetinib), schematized in *Figure 6—figure supplement 1C'*. *Figure 6—source data 1* presents raw data.

The online version of this article includes the following source data and figure supplement(s) for figure 6:

**Source data 1.** RNF43-overexpressing melanoma cells do not develop resistance to BRAF V600E targeted therapies.

**Figure supplement 1.** RNF43-overexpressing melanoma cells do not develop resistance to BRAF V600E targeted therapies.

**Figure supplement 1—source data 1.** RNF43-overexpressing melanoma cells do not develop resistance to BRAF V600E targeted therapies.

cells than the ones related to the WNT5A-specific events (*Arozarena et al., 2011*; *Arozarena and Wellbrock, 2019*; *Uka et al., 2020*; *Webster and Weeraratna, 2013*).

To strengthen our results, we exposed A375 and A2058 parental and RNF43-expressing cells temporarily and chronically to vemurafenib, MEK inhibitor – cobimetinib and to the FDA-approved combination of these drugs (*Ascierto et al., 2016*; *Signorelli and Shah Gandhi, 2017*; experimental scheme in *Figure 6—figure supplement 1C'*). We failed to obtain A375-resistant line, but A2058 acquired resistance to cobimetinib alone and its combination with vemurafenib (*Figure 6H and L*). Treatments and resistance were validated by pERK1/2 levels (*Figure 6H*). A2058 enriched with ectopically expressed RNF43 did not survive cobimetinib monotherapy and cobimetinib with vemurafenib combination, whereas vemurafenib-only treatment did not fully eliminate those cells (*Figure 6L*). Thus, we aimed to detect the differences in mechanisms behind targeted therapies resistance acquisition to characterize better the function of RNF43 in melanoma. Firstly, we noticed the shift of VANGL1 in all delivered resistance models, while VANGL2 was shifted only in cells being in the long-term culture with cobimetinib and with cobimetinib and vemurafenib (*Figure 6H*). This corresponds with decreased *RNF43* and *ZNRF3* expression (*Figure 6J*, *Figure 6—figure supplement 1D*) and increased *WNT5A* mRNA level (*Figure 6K*). Interestingly, both MITF protein level and *MITF* gene expression increased in VR cells and decreased in all cobimetinib-insensitive lines (*Figure 6H and I*). Moreover, *MITF* upregulation was accompanied by clear pigmentation of resistant cells (*Figure 6I'*), suggesting cell phenotype change (*Arozarena and Wellbrock, 2019*; *Ji et al., 2015*; *Müller et al., 2014*). Expression of *ROR1* decreased in case of MEKi resistance, while ROR2 remained unaffected (*Figure 6—figure supplement 1E and F*). Contrary to A375, *ROR1* did not significantly increase in VR-resistant cells and we did not detect the increased DVL2 and DVL3 phosphorylation (*Figure 6—figure supplement 1C*).

Altogether, these data confirm earlier findings on the importance of WNT5A signaling in the acquisition of resistance to targeted therapies and demonstrate that RNF43 can block this process. Moreover, we show that RNF43 could be a negative regulator of melanoma phenotype plasticity by targeting MITF$^{-low}$/WNT5A$^{-high}$ melanoma cells (*Hoek et al., 2006*; *Kim et al., 2017*; *Sensi et al., 2011*; *Tirosh et al., 2016*).

## RNF43 as onco-suppressor in vivo: impact on tumors and resistance to vemurafenib

Next, we aimed to confirm our results also in the in vivo model. We decided to use cell lines with the same origin but varying in the *RNF43* expression. For this purpose, A375 RNF43 TetON cells and A375 control lines (*Figure 4—figure supplement 3B and C*) were tested by qPCR. A375 control line had the lowest expression (referred to as '*RNF43 low*') of *RNF43*, and doxycycline-untreated A375 RNF43 TetON showed intermediate expression due to TetON leakage ('*RNF43 mid*') that could be further enhanced by the doxycycline treatment ('*RNF43 high*') (*Figure 7A*). Next, we prepared and tested a novel vemurafenib formulation in 25% Kolliphor-water solution ensuring the solubility of this BRAFi and which could be administrated by oral gavage in the previously published dose 25 mg/kg (*Figure 7—figure supplement 1A*; *Wang et al., 2019*). Cells were injected intradermally into the NOD-Rag1$^{null}$ IL2rg$^{null}$ mice. Animals were divided into treatment cohorts once tumors became prominent in size (*Figure 7B*, *Figure 7—figure supplement 1B*). Cohort I (*RNF43 low*) had significantly shorter survival and formed macroscopic tumors earlier than *RNF43 mid* and *high* cohorts (*Figure 7C*, *Figure 7—figure supplement 1C*). Moreover, tumors originating from cells with the lowest *RNF43* expression were significantly bigger than *RNF43 mid* ones (*Figure 7D*). To validate RNF43 expression induction and impact on its targets, protein lysates from tumors were analyzed by western blotting (*Figure 7—figure supplement 1D*). We found out that the level of RNF43-specific signal in cohorts I–III was significantly and inversely correlated with ROR1 and VANGL2 protein levels and similar trend was also noticed for DVL2 phosphorylation (*Figure 7E*, *Figure 7—figure supplement 1D and E*), recapitulating the described above in vitro effect also in the in vivo experiment. Importantly, we observed the RNF43 leakage (HA signal in *Figure 7—figure supplement 1D*), meaning that TetON system did not efficiently suppress the ectopic expression of RNF43 even in the absence of doxycycline, which also provides explanation for different *RNF43* levels in studied here cell lines (*Figure 7A*). In conclusion, the dose-dependent effect mediated by RNF43 showed the features of the onco-suppression in melanoma in vivo model.

Further, we also investigated the synergistic effect of RNF43 and vemurafenib in vivo. RNF43 mid and RNF43 high cohorts received vemurafenib, and this treatment significantly prolonged the survival (*Figure 7F*), validating our formulation and dosing efficacy. Importantly, doxycycline supplementation (*RNF43 high*) in addition to vemurafenib resulted in the more efficient suppression of tumor growth, suggesting the delayed the BRAFi resistance acquisition (*Figure 7G*). Nevertheless, there was no difference at the experimental end point (*Figure 7F and H*), possibly due to the loss of transgene expression observed in cells during this long-term experiment (HA signal in *Figure 7—figure supplement 1D*). Moreover, the protein level of WNT5A was significantly increased in tumors treated with vemurafenib, which could help to bypass the RNF43-related effect as well (*Figure 7—figure supplement 1E'''*). Our main findings are summarized as a graphical abstract (*Figure 8*).

## Discussion

Our study identified RNF43 as an inhibitor of noncanonical WNT5A-induced signaling. RNF43 physically interacted with multiple receptor components of the Wnt/PCP pathway such as ROR1/2, VANGL1/2, or DVL1/2/3 and triggered degradation of VANGL2 and membrane clearance of ROR1, ultimately resulting in the reduced cell sensitivity to WNT5A. The newly discovered RNF43 action in WNT5A-mediated signaling seems to be mechanistically different than the well-known function in the Wnt/β-catenin pathway. For example, we observed ROR1 and VANGL2 interaction with RNF43 in the absence of DVL. In contrast, DVL seems to be essential for the activity of RNF43 in the Wnt/β-catenin pathway (*Jiang et al., 2015*). Further, the inhibitory action of RNF43 in WNT5A signaling could not be blocked by inhibition of lysosomal pathway, in contrast to the earlier observations in WNT/β-catenin pathway (*Koo et al., 2012*). On the other side, WNT5A signaling can be, similarly to Wnt/β-catenin, promoted by RNF43 inhibitors from R-SPO family. Also, in line with the earlier findings that RNF43 leads to the packing of ubiquitinated FZD to the RAB5$^+$ endosomes (*Koo et al., 2012*), ROR1 is as well internalized via a clathrin-dependent mechanism into RAB5$^+$ and RAB11+ endosomes. Interestingly, ROR1 internalization is transient. We can speculate that it represents the first step in the silencing of the noncanonical Wnt pathway, which further translates into stable inhibition, for example, via

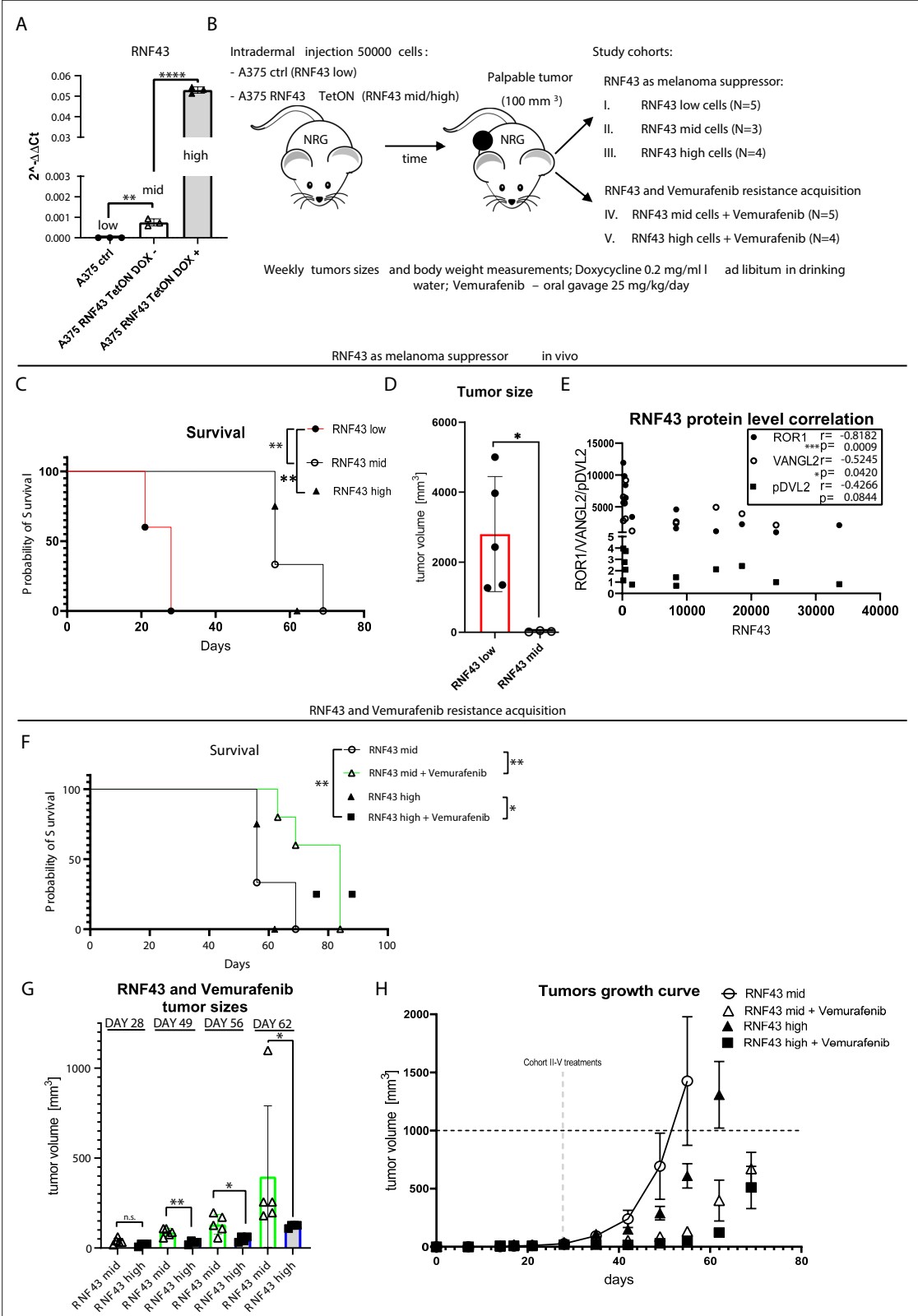

**Figure 7.** RNF43 inhibits melanoma proliferation and response to vemurafenib in vivo. (**A**) RT-qPCR results – expression of the *RNF43* gene in control (*RNF43* low) and A375 RNF43 TetON cells in the absence (*RNF43* mid) and presence of doxycycline (*RNF43* high). Results are presented as $2^{-\Delta\Delta Ct} \pm$ SD, two-tailed t-test: **p<0.01, ****p<0.0001, N = 3. Relative expression level was normalized to the *B2M* and *GAPDH* genes expression. (**B**) Schematic representation of the in vivo experiment based in the immunodeficient NOD-Rag1[null] IL2rg[null] mouse strain. A375 and its variants were injected

*Figure 7 continued on next page*

*Figure 7 continued*

intradermally (50,000 cells in PBS/mouse). Animals were divided into five cohorts once palpable tumors were formed. Ectopic RNF43 expression was induced by doxycycline presence in the drinking water. Vemurafenib was delivered daily by oral gavage as 25% Kolliphor in water formulation (see *Figure 7—figure supplement 1A*). Tumors sizes and animal weight were checked weekly. Mice were sacrificed when tumors reached approximately 1000 mm$^3$. (**C**) Survival within cohorts I–III. Injection with cells having the lowest *RNF43* expression led to the shortest mean survival (28 days). Experiment in cohorts II (56 days) and III (62 days) was significantly longer, Mantel–Cox test: *p<0.05, **p<0.01. (**D**) Cells expressing RNF43 higher than the endogenous level have impaired ability to proliferation in vivo. Tumor sizes at end point for the *RNF43* low cohort (N = 5) with RNF43 mid (N = 3), two-tailed t-test: *p=0.0297. (**E**) RNF43 negatively regulates its targets also in vivo. Correlation analysis of RNF43 western blot signals with ROR1, VANGL2 protein levels, and phosphorylation status of DVL2 (shown in *Figure 7—figure supplement 1D*). Results for cohorts I–III were analyzed (N = 12), one-tailed Spearman correlation test, p and r values are shown in the graph. (**F**) Survival analysis showing vemurafenib treatment efficacy. Doxycycline treatment for RNF43 induction did not increase further vemurafenib effect. Mean survival: RNF43 mid: 56 days; RNF43 mid + vemurafenib: 84 days; RNF43 high: 62 days; RNF43 high + vemurafenib: 76 days, Mantel–Cox test: *p<0.05, **p<0.01. (**G**) RNF43 delays acquisition of vemurafenib resistance in vivo. Cohort V (N = 4) receiving doxycycline for RNF43 overexpression induction and vemurafenib (RNF43 high + vemurafenib) had significantly smaller tumors at days 49–62 than cohort IV (RNF43 mid + vemurafenib), tumors were not different at the treatments starting point (day 28) two-tailed t-test: *p<0.05, **p<0.01. (**H**) Tumor growth curve – caliper measurements of tumors sizes within experimental groups II–V. Treatment starting points are marked and presented in *Figure 7—figure supplement 1B*. All used data are presented in *Figure 7—source data 1*.

The online version of this article includes the following source data and figure supplement(s) for figure 7:

**Source data 1.** RNF43 inhibits melanoma proliferation and response to vemurafenib in vivo.

**Figure supplement 1.** RNF43 inhibits melanoma proliferation and response to vemurafenib in vivo.

**Figure supplement 1—source data 1.** RNF43 inhibits melanoma proliferation and response to vemurafenib in vivo.

VANGL2 sequestration. It remains to be studied how RNF43 in a coordinated manner controls both WNT/β-catenin and noncanonical WNT pathways.

We demonstrate that the newly characterized RNF43-WNT5A regulatory module controls WNT5A signaling and biology in melanoma. WNT5A-induced signaling plays a crucial role in this cancer type. Up to date, 5 -year survival of metastatic melanoma patients rates between 5 and 19%, depending on the location and the number of metastases (*Sandru et al., 2014*). Elevated expression of *WNT5A* associates with negative overall survival in melanoma (*Da Forno et al., 2008*; *Luo et al., 2020*; *Weeraratna et al., 2002*); we have observed an inverse correlation for *RNF43*, which was a positive prognostic factor in melanoma and got silenced as melanoma progressed. WNT5A promotes multiple proinvasive features of melanoma cells such as EMT-like process, invasion, metastasis, cell proliferation, and ECM remodeling by melanoma cells (*Dissanayake et al., 2008*; *Dissanayake et al., 2007*; *Fernández et al., 2016*; *Lai et al., 2012*). Moreover, WNT5A ligand co-receptors – ROR1 and ROR2 – have been already described in melanoma as key factors driving invasion in vitro and in vivo (*Fernández et al., 2016*; *Lai et al., 2012*; *O'Connell et al., 2013*; *O'Connell et al., 2010*). Ultimately, RNF43 overexpression here efficiently suppressed all tested pro-metastatic properties of melanoma cells associated with WNT5A and its (co)receptors. Among those, the clinically most relevant is the acquisition of resistance to BRAF inhibitor vemurafenib (*Chapman et al., 2011*; *Flaherty et al., 2010*).

*BRAF V600E* mutation appears in up to 50% of melanoma cases, which results in the oncogenic activation of MAPK pathway (*Akbani et al., 2015*; *Wan et al., 2004*). Vemurafenib (PLX4032), a compound selectively inhibiting BRAF V600E, showed positive clinical effects in melanoma (*Bollag et al., 2012*; *Joseph et al., 2010*). Unfortunately, most of the patients develop resistance to vemurafenib treatment and progress (*Chapman et al., 2011*). Multiple mechanisms underlying acquisition of resistance were described (*Arozarena and Wellbrock, 2019*; *Arozarena and Wellbrock, 2017a*; *Johnson et al., 2015*; *Luebker and Koepsell, 2019*; *Schmitt et al., 2019*; *Su et al., 2020*; *Su et al., 2017*; *Talebi et al., 2018*; *Tirosh et al., 2016*). Among those mechanisms, WNT5A signaling has a prominent role – *WNT5A* expression was shown to positively correlate with vemurafenib resistance (*Anastas et al., 2014*; *Prasad et al., 2015*; *Webster et al., 2015*) and WNT5A treatment decreased melanoma cells' response to the vemurafenib (*Anastas et al., 2014*; *O'Connell et al., 2013*). Our finding that RNF43-controlled regulatory axis could completely block the development of resistance to BRAF and also MEK inhibition further highlights the importance of WNT5A signaling in this process and uncovers a mechanism that can be explored therapeutically.

Previous studies showed that melanoma displays remarkable phenotypic plasticity upon targeted therapy (*Hoek et al., 2006*; *Hoek and Goding, 2010*; *Kemper et al., 2014*; *Rambow et al., 2018*). This is also well demonstrated in our A2058 model where vemurafenib-resistant cells become

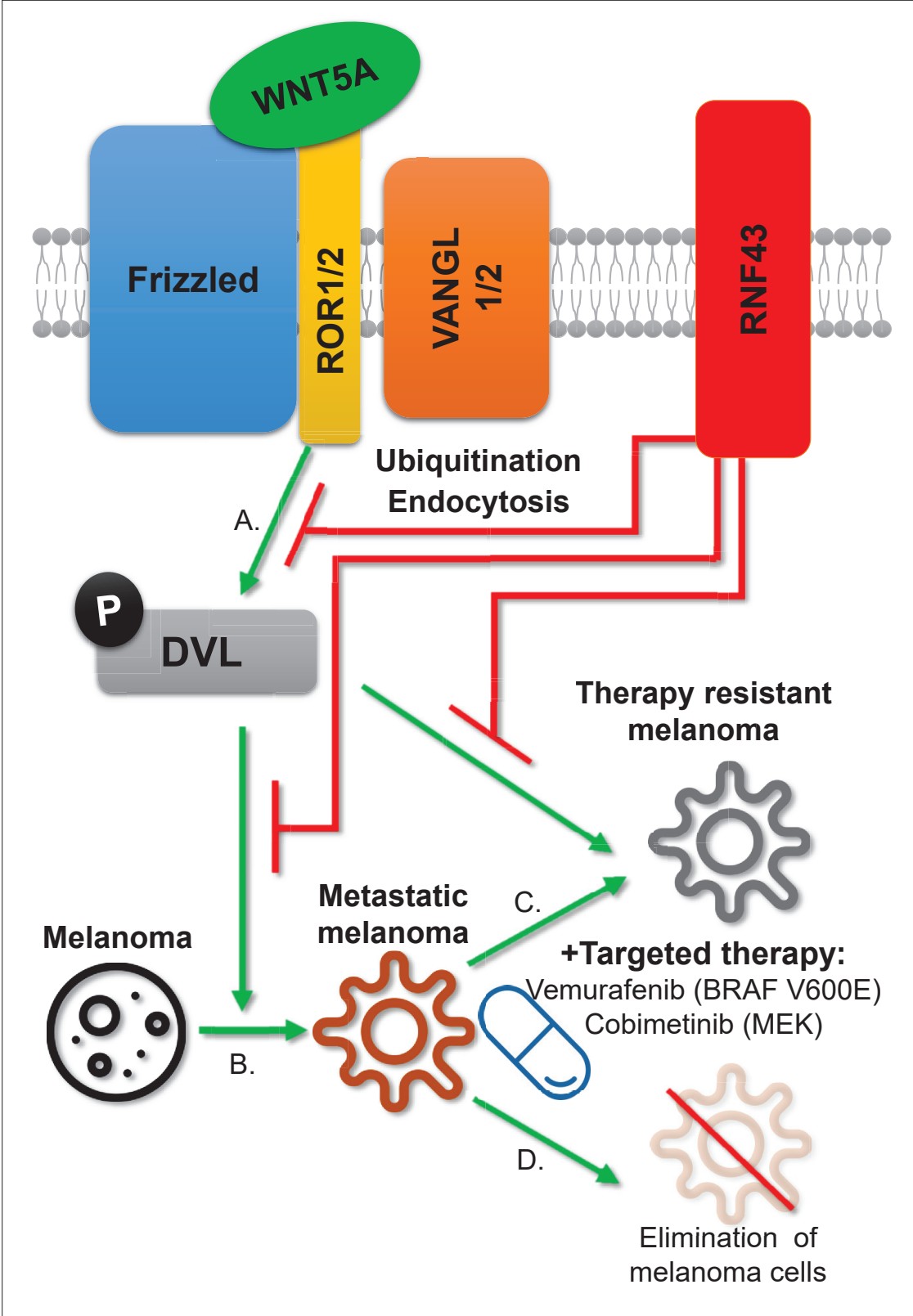

**Figure 8.** RNF43 inhibits WNT5A-driven signaling and suppresses melanoma invasion and resistance to the targeted therapy. Graphical summary. RNF43 is an inhibitor of the noncanonical WNT5A-induced pathway. RNF43 interacts with receptor complexes of the Wnt/PCP signaling and its enzymatic activity results in the reduced cells sensitivity to WNT5A (**A**). In melanoma, WNT5A promotes invasion and metastasis (**B**) as well as resistance to targeted therapies, including treatments with vemurafenib – inhibitor of commonly mutated BRAF kinase and cobimetinib-targeting activity of the

*Figure 8 continued on next page*

*Figure 8 continued*

MEK enzyme (**C**). RNF43 blocks melanoma-invasive properties via interference with the nonconical Wnt pathway, leading to the increased sensitivity to treatment (**D**).

melanotic (MITF$^{high}$), whereas cobimetinib- and especially vemurafenib/cobimetinib-double-resistant cells have MITF$^{low}$/WNT5A$^{high}$ phenotype that is characteristic for highly invasive melanoma with the dedifferentiated phenotype (*Ahn et al., 2017*; *Anastas et al., 2014*; *Arozarena and Wellbrock, 2019*; *Massi et al., 2020*; *Webster et al., 2015*). Inhibition of the WNT5A pathway by RNF43 could block one of the trajectories of resistance acquisition by Darwinian selection of preexisting subpopulation (*Chisholm et al., 2015*), thereby promoting less metastatic phenotype (*Bai et al., 2019*). This is supported by our observation that vemurafenib-treated A2058 cells can partially overcome RNF43 OE by phenotypic switch to MITF$^{high}$ phenotype, which is not the case in cobimetinib-treated cells. From the other angle, the low *RNF43* expression might be explored as a marker of resistant melanoma phenotype. *RNF43* is a β-catenin target gene (*Takahashi et al., 2014*), which could be a reason for the low *RNF43* levels in metastatic melanomas where Wnt/β-catenin signaling is inhibited (*Arozarena et al., 2011*; *Kageshita et al., 2001*; *Uka et al., 2020*). Altogether, we provide evidence that RNF43 can act as a tumor suppressor and a negative regulator of the acquisition of the resistance to the targeted therapy.

The relevance of our findings is likely not limited to melanoma. Signaling cascade RSPO–LGR4/5–RNRF43/ZNRF3 has been shown to regulate a variety of biological processes. In light of our results, it is tempting to speculate that WNT5A-RNF43 axis regulates other developmental, physiological, and pathophysiological conditions. For example, *WNT5A* is overexpressed in gastric cancer where it positively correlates with the presence of lymph node metastasis, tumor depth, EMT induction, and poor prognosis (*Astudillo, 2020*; *Hanaki et al., 2012*; *Kanzawa et al., 2013*; *Kurayoshi et al., 2006*; *Nam et al., 2017*; *Saitoh et al., 2002*). Notably, reduced RNF43 function is a negative prognosis factor in gastric cancer patients (*Gao et al., 2017*; *Neumeyer et al., 2019a*; *Niu et al., 2015*) and RNF43 loss-of-function type of mutation exacerbated *Helicobacter pylori*-induced gastric tumor carcinogenesis associated with the upregulation of *WNT5A* mRNA level (*Katoh, 2007*; *Li et al., 2014*; *Neumeyer et al., 2019b*; *Peek and Crabtree, 2006*). Further, in colorectal cancer, RNF43 mutations were found to associate with *BRAF* V600E mutation (*Matsumoto et al., 2020*; *Yan et al., 2017*). These results suggest the existence of a more universal functional WNT5A-RNF43 axis where RNF43 acts as a gate-keeper guarding the abnormal pro-cancerogenic noncanonical Wnt pathway activation.

Further exciting avenues relate to the importance of RSPO-RNF43/ZNRF3 module in the regulation of multiple developmental processes dependent on WNT5A. There are literature hints that suggest that indeed WNT5A signaling is fine-tuned by RNF43/ZNRF3 during convergent extension movements. The regulation of Rspo3 has been proven in *Xenopus* embryogenesis, where it regulates gastrulation movements and head cartilage morphogenesis in a manner involving Wnt5a and Syndecan-4 binding by R-spondin. Strikingly, *Rspo3* antisense morpholino caused a phenotype characteristic for the noncanonical Wnt signaling pathway – *spina bifida* (*Ohkawara et al., 2011*). Similarly, overexpression of *Znrf3* in zebrafish embryos caused shortened body axis and abnormal shape of somites, phenotypes also recognized as typical for Wnt/PCP pathway perturbances (*Hao et al., 2012*). And, finally in mammals, a fraction of *Znrf3* KO mice showed an open neural tube phenotype (*Hao et al., 2012*), again reminiscent of defective Wnt/PCP signaling. Altogether, these observations, together with our data, suggest that RSPO-RNF43/ZNRF3 signaling represents an evolutionary conserved and widely used mechanism used to control the activation of noncanonical WNT signaling.

## Materials and methods
### 1. Cell lines and treatments

T-REx-293 (R71007, Thermo Fisher Scientific), GFP labeled human melanoma A375 wild-type (A375) and its metastatic derivates A375 IV (*Kucerova et al., 2014*), A2058 (ECACC 91100402), and MelJuso (*Štětková et al., 2020*) (kindly gifted by Stjepan Uldrijan) cell lines were propagated in the Dulbecco's modified Eagle's medium (DMEM, 41966-029, Gibco, Life Technologies) supplemented with the 10% fetal bovine serum (FBS, 10270-106, Gibco, Life Technologies), 2 mM L-glutamine (25030024, Life Technologies), 1% penicillin-streptomycin (XC-A4122/100, Biosera) under 5% (vol/vol) $CO_2$ controlled

atmosphere at 37 °C. Routine checks for mycoplasma contamination were performed. For inhibition of endogenous Wnt ligands, cells were treated with the 0.5 µM Porcupine inhibitors C-59 (ab142216, Abcam) or 1 µM LGK-974 (1241454, PeproTech). The time points and doses have been chosen based on the purpose of the experiment. Changes in the phosphorylation of the WNT5A-downstream proteins in the noncanonical Wnt pathway transduction have been analyzed after 3 hr with the recombinant human WNT5A (645-WN, R&D Systems) in doses 40–100 ng/ml as given in the figure legends. Longer stimulation (overnight and longer) has been used in the functional experiments. For canonical Wnt signaling activation, the recombinant human WNT3A (5036-WN, R&D Systems) was used overnight in 40 ng/ml, 60 ng/ml, or 80 ng/ml concentrations. Co-treatment with the recombinant human R-Sponidin-1 (120-38, PeproTech) in 50 ng/ml dose was applied where indicated. Dansylcadaverine (D4008, Sigma-Aldrich) 50 µM treatment along with 3 hr tetracycline was applied to block clathrin-dependent endocytosis pathway (**Blitzer and Nusse, 2006**).

For preparation of stable cell lines, antibiotic selection after plasmid DNA transfection was performed using 5 µg/ml blasticidin S (3513-03-9, Santa Cruz Biotechnology) or 200 µg/ml of hygromycin B (31282-04-9, Santa Cruz Biotechnology) for T-REx-293 cells and accordingly 400 µg/ml and 5 µg/ml in case of A375 melanoma cell line. As a result, tetracycline-inducible T-REx-293 RNF43 and RNF43 Mut1 TetON, T-REx-293 *RNF43/ZNRF3* dKO RNF43 TetON, A375+ RNF43, and A375 IV + RNF43 were obtained. A2058 RNF43 TetON, MelJuso RNF43 TetON A375 RNF43 TetON, and A375 TetON ctrl (not expressing exogenous RNF43) cells were obtained by lentiviral transduction of doxycycline-inducible pCW57-RNF43 (blast) plasmid encoding HA- and FLAG- tagged RNF43, using published protocol (**Barta et al., 2016**), followed by 5 µg/ml blasticidin S selection and limiting dilution.

T-REx-293 *DVL1/2/3* tKO cells were described previously (**Paclíková et al., 2017**). For transgene expression induction (TetON), T-REx-293 cells were treated with 1 µg/ml of tetracycline (60-54-8, Santa Cruz Biotechnology) for the indicated time (3 hr to overnight) and melanoma cells were induced overnight by 1 µg/ml doxycycline (HY-N0565B, MedChem Express). Lysosomal degradation pathway was blocked by the 10 µM chloroquine (C662, Sigma) treatment, whereas 10 µM MG-132 (C2211, Sigma) was used for the proteasome inhibition. Generation of the melanoma cells resistant to vemurafenib (HY-12057, MedChem Express) and cobimetinib (HY-13064, MedChem Express) was performed according to the published protocols (**Anastas et al., 2014**). Resistant cells were cultured in the presence of 2 µM (A375) and 5 µM (A2058) of vemurafenib and 0.5 µM cobimetinib or their combination (A2058). For transient treatments (24 hr) of melanoma cell lines, 0.5 µM vemurafenib has been used (A375) or 2.5 µM of vemurafenib and 0.5 µM of cobimetinib (A2058).

## 2. Plasmids/cloning

Backbone of the plasmid pcDNA4-TO-RNF43-2xHA-2xFLAG (kindly gifted by Bon-Kyoung Koo together with pcDNA4-TO-RNF43Mut1-2xHA-2xFLAG; **Koo et al., 2012**) was used for further cloning. Briefly, for generation of the BioID-inducible pcDNA4-TO-RNF43-BirA*-HA plasmid, cDNA encoding RNF43 without stop codon was amplified by the PCR and cloned into the pcDNA3.1 MCS-BirA(R118G)-HA (Addgene plasmid #36047; RRID:Addgene_36047) using HpaI (ER1031, Thermo Fisher Scientific) and EcoRI (ER0271, Thermo Fisher Scientific) restriction enzymes to fuse it in frame with the BirA*-HA sequence. Then, RNF43-BirA*-HA cDNA was amplified and cloned by the In-Fusion cloning method (639690, Takara Bio) into linearized by HindIII (ER0501, Thermo Fisher Scientific) and XbaI (ER0681, Thermo Fisher Scientific) pcDNA4-TO plasmid. To eliminate BirA* enzyme-mediated potential false-positive results, pcDNA3-RNF43-HA was prepared by subcloning RNF43 PCR product containing HA encoding sequence in the reverse primer to the pcDNA3 backbone (Invitrogen). pCW57-RNF43 (blast) plasmid was obtained by RNF43-HA-FLAG cDNA cloning into the EcoRI site of the pCW57-MCS1-P2A-MCS2 (Blast) backbone (Addgene #80921; RRID:Addgene_80921) by the In-Fusion cloning method. All obtained plasmids were verified by the Sanger sequencing method.

Other plasmids used were described previously and included myc-Vangl1, GFP-Vangl2, GFP-Vangl2ΔN, GFP-Vangl2ΔC, GFP-Vangl2ΔNΔC (**Belotti et al., 2012**), pLAMP1-mCherry (Addgene #45147), pEGFP-C1-Rab5a (**Chen et al., 2009**), GFP-rab11 WT (Addgene #12674), His-ubiquitin (**Tauriello et al., 2010**), pcDNA3-Flag-mDvl1 (**Tauriello et al., 2010**), pCMV5-3xFlag Dvl2 (Addgene #24802), pCDNA3.1-Flag-hDvl3 (**Angers et al., 2006**), pcDNA3.1-hROR1-V5-His (gifted by Kateřina Tmějová), pcDNA3-Ror2-Flag and pcDNA3-Ror2-dCRD-FLAG (**Sammar et al., 2004**), pRRL2_ROR1ΔCYTO and

**Table 1.** Cloning and mutagenesis primers.

| Primer | Sequence | Purpose |
| --- | --- | --- |
| RNF43 BirA*F | ATGCAGTTAACATGAGTGGTGGCCACCAGCTG | RNF43 cDNA cloning into pcDNA3.1 MCS-BirA(R118G)-HA |
| RNF43 BirA*R | ATGCAGAATTCCACAGCCTGTTCACACAGCTCCT | |
| RNF43 InFusion F | GTTTAAACTTAAGCTTATGAGTGGTGGCCACCAG | RNF43-BirA(R118G)-HA into pcDNA4 |
| RNF43 InFusion R | AAACGGGCCCTCTAGACTATGCGTAATCCGGTACA | |
| RNF43-HA F | TTAAAGCTTATGAGTGGTGGCCACCAG | |
| RNF43-HA R | ATCGATATCTCAAGCGTAATCTGGAACATCGTATGGG TACACAGCCTGTTCACACAGCT | RNF43-HA cloning into pcDNA3 |
| pCW57-RNF43 InFusion F | ATTGGCTAGCGAATTATGAGTGGTGGCCACCAGC | |
| pCW57-RNF43 InFusion R | CGGTGTCGACGAATTTCAGGCGTAGTCGGGCACG | pCW57-RNF43 generation |

pRRL2_ROR1ΔTail (*Gentile et al., 2011*), hCas9 (Addgene #41815), gRNA_GFP-T1 (Addgene #41819), and PiggyBack-Hygro and Transposase coding plasmids (gifted by Bon-Kyoung Koo). Sequences of primers used for cloning are presented in *Table 1*.

## 3. CRISPR/Cas9

For targeting *RNF43* and *ZNRF3* in the T-Rex-293, gRNAs *TGAGTTCCATCGTAACTGTGT*TGG (PAM) and AGACCCGCTCAAGAGGCCGGTGG were cloned into gRNA_GFP-T1 backbone and transfected together with PiggyBack-Hygro and Transposase coding plasmids using polyethylenimine (PEI) in a way described below. For *ROR1* and *WNT5A* knockout cell lines generation, gRNA *CCATCTATG-GCTCTCGGCTG*CGG (ROR1) and AGTATCAATTCCGACATCGAAGG (WNT5A) were used. Transfected cells were hygromycine B selected and seeded as single cells. Genomic DNA isolation was performed using DirectPCR Lysis Reagent (Cell) (Viagen Biotech), Proteinase K (EO0491, Thermo Fisher Scientific), and DreamTaq DNA Polymerase (EP0701, Thermo Fisher Scientific) according to the manufacturer's instructions. PCR products were analyzed by restriction digestion using Taal (ER1361, Thermo Fisher Scientific) in case of *RNF43*, HpaII (ER0511, Thermo Fisher Scientific) – *ZNRF3*, TaqI (ER0671, Thermo Fisher Scientific) – *WNT5A* and TseI (R0591S, New England BioLabs) – *ROR1* for detection of Cas9-mediated disruptions in the recognition sites.

For targeting *RNF43/ZNRF3* in the A375 and in the A375 IV melanoma lines, gRNAs *AGTTAC-GATGGAACTCA*TGG (RNF43) and CTCCAGACAGATGGCACAGTCGG (ZNRF3) were accordingly cloned by described protocol into the pU6-(BbsI)CBh-Cas9-T2A-mCherry (Addgene #64324) and pSpCas9(BB)-2A-GFP (PX458) (Addgene #48138) backbones, transfected and sorted as single, GFP, and mCherry double-positive cells. These were then analyzed by restriction enzymes Hin1II (ER1831, Thermo Fisher Scientific) and Taal as described above. Finally, PCR products were sequenced using the Illumina platform and compared with the reference sequence (*Malcikova et al., 2015*). Sequencing results are presented in *Supplementary file 1*.

## 4. RNF43 BioID analysis

Data are available via ProteomeXchange (*Deutsch et al., 2020*) with identifier PXD020478 in the PRIDE database (*Perez-Riverol et al., 2019*). The analysis of the mass spectrometric RAW data files was carried out using the MaxQuant software (version 1.6.2.10) using default settings unless otherwise noted. MS/MS ion searches were done against modified cRAP database (based on http://www.thegpm.org/crap) containing protein contaminants like keratin, trypsin, etc., and UniProtKB protein database for *Homo sapiens* (ftp://ftp.uniprot.org/pub/databases/uniprot/current_release/knowledge-base/reference_proteomes/Eukaryota/UP000005640_9606.fasta.gz; downloaded 19.8.2018, version 2018/08, number of protein sequences 21,053). Oxidation of methionine and proline, deamidation (N, Q) and acetylation (protein N-terminus) as optional modification, carbamidomethylation (C) as fixed modification, and trypsin/P enzyme with two allowed miss cleavages was set. Peptides and proteins

with FDR threshold <0.01 and proteins having at least one unique or razor peptide were considered only. Match between runs was set among all analyzed samples. Protein abundance was assessed using protein intensities calculated by MaxQuant. Protein intensities reported in proteinGroups.txt file (output of MaxQuant) were further processed using the software container environment (https://github.com/OmicsWorkflows, *Kristina, 2021*), version 3.7.2 a. Processing workflow is available upon request. Briefly, it covered (1) removal of decoy hits and contaminant protein groups, (2) protein group intensities log2 transformation, (3) LoessF normalization, (4) imputation by the global minimum, and (5) differential expression using LIMMA statistical test. Prior to volcano plot plotting, suspected BirA* binders were filtered out (proteins identified by at least two peptides in both technical replicates of particular BirA* sample, and present in more than three samples). Volcano plot was created in R using ggplot2 and ggrepel R packages by R version 3.6.1. Proteins with an adjusted p-value < 0.05 and log fold change >1 were further subjected to gene ontology tools, considering only the first ID of majority protein IDs: g:Profiler online tool (https://biit.cs.ut.ee/gprofiler/gost, version e98_eg45_p14_ce5b097; *Raudvere et al., 2019*) was used and selected GO terms were highlighted. RNF43 interactors from BioID assay are listed in *Figure 1—source data 1*, and the results obtained by g:Profiler are presented in *Figure 1—source data 2*.

## 5. Transfection

T-REx-293 cells were transected using 1 µg/ml, pH 7.4 PEI, and plasmid DNA in a 4:1 ratio (*Paclíková et al., 2017*). Plasmid DNA were in amount of 3 µg for 6 cm culture dish (ubiquitination assay) and 6 µg for 10 cm dish (co-immunoprecipitation or stable cell lines preparation). Approximately 1 × 10^6 of A375 and A375 IV cells were electroporated with 6 µg of plasmid DNA utilizing Neon Transfection System (Thermo Fisher Scientific) 1200 V, 40 ms, 1 pulse. Culture media were changed 6 hr post-transfection.

## 6. His-ubiquitin pulldown assay

Cells were transfected with the plasmid encoding polyhistidine-tagged ubiquitin, RNF43-HA, or enzymatically inactive RNF43, protein of interest, and cultured overnight. Next, cells were treated with 0.2 µM epoxomicin (E3652, Sigma) for 4 hr and lysed in the buffer containing 6 M guanidine hydrochloride (G3272, Sigma), 0.1 M Na$_x$H$_x$PO$_4$ pH 8.0, and 10 mM imidazole (I5513, Sigma), sonicated, and boiled. Insoluble fraction was removed by the centrifugation (16,000 g, room temperature [RT], 10 min). For the pull down of tagged proteins, 10 µl of equilibrated in lysis buffer His Mag Sepharose beads Ni (GE28-9799-17, GE Healthcare) was added to each sample and kept on a roller overnight. Then, the beads were washed three times in the buffer containing 8 M urea (U5378, Sigma), 0.1 M Na$_x$H$_x$PO$_4$ pH 6.3, 0.01 M Tris, and 15 mM imidazole, resuspended in 100 µl of western blot sample buffer, boiled for 5 min, and loaded onto SDS-PAGE gel. Approximately 10% of cellular lysate was used as a transfection control after ethanol precipitation and resuspension in the western blot sample buffer.

## 7. Western blotting and antibodies

Western blot analysis was performed as described before using samples with the same protein amount, measured by the DC Protein Assay (5000111, Bio-Rad), or lysed directly in the sample buffer (2% SDS, 10% glycerol, 5% β-mercaptoethanol, 0.002% bromophenol blue, and 0.06 M Tris HCl, pH 6.8) and protease inhibitor cocktail (11836145001, Roche) after PBS wash (*Mentink et al., 2018*). Protein extraction from mouse tissues was done by homogenizing in the 1% SDS, 100 mM NaCl, 100 mM Tris, pH 7.4 buffer, sonication, clarification by centrifugation (16,000 g, 4 °C, 15 min), and protein concentration measurement. Next, volumes of samples containing the same protein amounts were mixed with western blot sampling buffer and loaded onto SDS-PAGE gels. Briefly, after electrophoretic separation, proteins were transferred onto Immobilon-P PVDF Membrane (IPVH00010, Millipore) and detected using primary and corresponding HRP-conjugated secondary antibodies on Fusion SL imaging system (Vibler) using Immobilon Western Chemiluminescent HRP Substrate (Merck, WBKLS0500). Molecular size of the bands is marked in each panel (kDa). A list of used antibodies is presented in Appendix 1—key resources table. Shifts of ROR1, ROR2, DVL2, DVL3, VANGL1, and VANGL2 are marked by arrowheads. Empty arrowhead marks phosphorylation-dependent shift. Densitometric analysis of western blot signals was performed using ImageJ software. Activation level

of DVL2 and DVL3 (*Bryja et al., 2007a*) is presented as the ratio of intensities of upper band; representing active – phosphorylated protein fraction and lower band (black arrowhead; unphosphorylated). Total DVL2 and DVL3 levels were quantified as the sum of two bands intensities.

## 8. Immunofluorescence and confocal microscopy

Cells growing on the glass were fixed in 4% paraformaldehyde (PFA) in PBS. Fixed cells were permeabilized by 0.1% Triton X-100 in PBS and blocked in 1% solution of bovine serum albumin (BSA) in PBS. Then, samples were incubated overnight at 4 °C with primary antibodies diluted in 1% BSA in PBS and washed. Corresponding Alexa Fluor secondary antibodies (Invitrogen) were incubated with samples for 1 hr at RT, along with 1 µg/ml Hoechst 33342 (H1399, Thermo Fisher Scientific) for nuclei staining. After PBS washes, the samples were mounted in the DAKO mounting medium (S3023, DAKO). Images were taken on the confocal laser scanning microscopy platform Leica TCS SP8 (Leica). For co-localization analysis, histograms for each channel were prepared in LAS X Life Science (Leica) software and plotted in GraphPad Prism 8. Co-localization is marked by arrowheads.

## 9. Immunoprecipitation

T-REx-293 cells were transfected with the proper plasmid DNA and cultured for 24 hr. Then, cells were washed two times with PBS and lysed for 15 min in the buffer containing 50 mM Tris pH7.6, 200 mM NaCl, 1 mM EDTA, 0.5% NP40, fresh 0.1 mM DTT (E3876, Sigma) and protease inhibitor cocktail (04693159001, Roche). Insoluble fraction was removed by centrifugation (16,000 g, RT, 15 min), 10% of total cell lysate was kept as western blot control. Lysates were incubated with 1 µg of antibody for 16 hr at 4 °C on the head-over-tail rotator. Next, 20 µl of protein G-Sepharose beads (17-0618-05; GE Healthcare) equilibrated in complete lysis buffer were added to each sample and incubated for 4 hr at 4 °C, following six washes using lysis buffer and resuspension in 100 µl of western blot sample buffer. Immunoprecipitation experiments were analyzed by the western blot.

## 10. Flow cytometric determination of ROR1 surface expression

Determination of the ROR1 surface expression of T-REx-293 and its derivates was done using anti-ROR1-APC (#130-119-860, Miltenyi Biotec) and Accuri C6 (BD Biosciences) (*RNF43/ZNRF3* dKO cells) or using BD FACSVerse Flow Cytometer (BD Biosciences) (TetON cells). Cells were harvested in 0.5 mM EDTA/PBS, washed in PBS, and incubated in 2% FBS in PBS with anti-ROR1-APC antibody (1:25, #130-119-860, Miltenyi Biotec) on ice for 30 min. The cells were washed and resuspended in PBS, and incubated with propidium iodide (10 ng/ml, #81845, Sigma-Aldrich) for 5 min to exclude dead cells from analysis. For the detection of ROR1 surface expression in HA-positive cells, ROR1-APC-stained cells were washed in PBS, fixed in 4% PFA at RT for 15 min, permeabilized in 0.02% Triton X-100 at RT for 15 min, and incubated with anti-HA antibody (1:1000, #9110, Abcam) in staining buffer at RT for 30 min. After two washes, cells were incubated with secondary antibody ALEXA Fluor 488 Donkey anti-Rabbit (#A21206, Invitrogen) at RT for 20 min, washed, and measured using FACS Verse (BD Biosciences). Data were analyzed using NovoExpress Software (ACEA Biosciences).

## 11. Quantitative polymerase chain reaction (qPCR)

Messenger RNA was isolated using the RNeasy Mini Kit (74106; Qiagen) according to the manufacturer's instructions. 1 µg of mRNA was transcribed to cDNA by the RevertAid Reverse Transcriptase (EP0442, Thermo Fisher Scientific) and analyzed by use of LightCycler 480 SYBR Green I Master (04887352001, Roche) and LightCycler LC480 (Roche). Results are presented as $2^{-\Delta\Delta CT}$ and compared by unpaired Student's t-test. Mean expression of *B2M* and *GAPDH* or *HSPCB* and *RPS13* (A375 VR cells) was used as reference. Primers are listed in Appendix 1—key resources table.

## 12. Databases

RNF43, VANGL1, and DVL3 gene expression in different melanoma stages was analyzed through Oncomine (RRID:SCR_007834; *Rhodes et al., 2004*) database in the different datasets (*Talantov et al., 2005*, *Xu et al., 2008*, *Haqq et al., 2005*). OncoLnc (*Anaya, 2016*) database was employed to elucidate whether the expression of the *RNF43, ZNRF3, VANGL1,* and *DVL3* gene expression has a significant impact on the melanoma patients' overall survival. RNF43 BioID data are available via

ProteomeExchange (RRID:SCR_004055; *Deutsch et al., 2020*) in the PRIDE database (PXD020478) (RRID:SCR_003411; *Perez-Riverol et al., 2019*).

## 13. Wound healing assay, Matrigel invasion assay, fluorescent gelatin degradation assay, invadopodia formation assay, and collagen I hydrogel 3D invasion assay

For the determination of cellular motility and invasive properties in vitro wound healing (*O'Connell et al., 2008*), Matrigel invasion towards 20 % FBS as chemoattractant followed by crystal violet staining of invaded cells, fluorescent gelatin degradation in the presence of 5% FBS after overnight starvation, and invadopodia formation assays were prepared according to the established protocols (*Makowiecka et al., 2016*). The wound gap was photographed using the Olympus ix51 inverted fluorescence microscope after 48 hr from scratch. Percentage of the cell-free surface was measured by ImageJ software. For the fluorescent gelatin degradation assay purpose, 80 ng/ml of rhWNT5A was used during 16 hr of cells' incubation on the coverslips coated with gelatin-Oregon Green conjugate (G13186, Thermo Fisher Scientific). Alexa Fluor 594 phalloidin (A12381, Thermo Fisher Scientific) and TO-PRO-3 Iodide (642/661) were employed for the cells' visualization on confocal microscopy platform Leica TCS SP8. For the invadopodia formation assay, an immunofluorescence imaging protocol employing phalloidin and anti-cortactin antibody was performed. Invadopodia – as structures double positive for F-actin and cortactin staining – was quantified for tested cell lines and conditions and presented as the number of invadopodia per one cell. Two independent repetitions were performed.

Collagen I hydrogel 3D invasion assay is a modification of the inverted vertical invasion assay (*McArdle et al., 2016*). Cells were plated on the μ-Slide 8 Well glass bottom coverslips (80827, Ibidi). At 80% confluence, the full medium was replaced with one containing 0.5% FBS for proliferation suppression. Doxycycline for RNF43 induction was applied at this step when needed. Next day, a solution of rat tail collagen type I in final concentration 1.5 mg/ml prepared accordingly to the manufacturer's protocol was overlaid over the cells and left for polymerization for 30 min at 37 °C, 5% $CO_2$. Then medium with final FBS concentration 10% and 100 ng/ml CXCL12 (350-NS, R&D Systems) or 100 ng/ml CCL21 (366-6C , R&D Systems) was added to the wells. After 24 hr, cells were PFA fixed, permeabilized, and stained with Hoechst 33342 (H1399, Thermo Fisher Scientific) and Alexa Fluor 594 phalloidin. Corresponding photos at the coverslip level and at 50 μm (A375 and A2058) or 70 μm (A375 IV) were taken using a confocal laser scanning microscopy platform Leica TCS SP8 (Leica). Invasion index was calculated as the ratio of invaded cells at a specified height to the number of noninvasive ones.

## 14. Colony formation assay

To assess the ability of colony formation in the presence of 0.3 μM vemurafenib, 300 of the melanoma cells were plated onto 24-well plate and were subsequently cultured for 7 days. After that time, the medium was removed and colonies were washed in PBS, fixed in the ice-cold methanol for 30 min, and stained with 0.5% crystal violet in 25% methanol. After washing and drying, bound crystal violate was eluted with 10% acetic acid and absorbance at 590 nm was measured on Tecan Sunrise plate reader. Results were normalized to the nontreated A375 wild-type results.

## 15. Animal studies

Animal experiments were approved by the Academy of Sciences of the Czech Republic (AVCR 85/2018), supervised by the local ethical committee, and performed by certified individuals (Karel Souček, Markéta Pícková, Ráchel Víchová).

A375 ctrl and A375 RNF43 TetON cells were implanted as a suspension of 50,000 cells in 50 μl saline intradermally into 9 -week-old males of NOD-Rag1[null] IL2rg[null] strain, obtained from Jackson Laboratory. Animals were checked daily, and weight and tumors sizes were measured weekly along perpendicular axes using an external caliper. Tumor volumes were calculated using the equation volume = ½ (length × width$^2$). Upon tumor establishment, animals were divided into five cohorts based on administered cells and treatment: (I) A375 ctrl + vemurafenib (N = 5); (II) A375 RNF43 TetON Dox - (N = 3); (III) A375 RNF43 TetON Dox + (N = 4); (IV) A375 RNF43 TetON VEMURAFENIB Dox - (N = 5); and (V) A375 RNF43 TetON VEMURAFENIB Dox + (N = 4). Doxycycline supplemented for RNF43 expression induction was administrated in drinking water, 0.2 mg/ml in 0.1% weight:volume sucrose. Control cohort obtained drinking water with 0.1% sucrose. Vemurafenib was supplemented daily by

oral gavage in concentration of 25 mg/kg/day as freshly prepared formulation in 25% Kolliphor ELP (61791-12-6, Sigma-Aldrich) with 2.5% DMSO. Animals not treated with vemurafenib received the same formulation without inhibitor. Mice were sacrificed when tumors reached approximately 1000 mm$^3$. Tumor samples were analyzed by western blot, and the times to reach the experimental end point was compared.

## 16. Software and statistics

Statistical significance was confirmed by two-tailed paired or unpaired Student's t-tests. Survival was analyzed by Mantel–Cox test. Correlation between RNF43(HA) protein level and its targets in the in vivo experiments was tested by one-tailed Spearman correlation test. Statistical significance levels were defined as *p<0.05, **p<0.01, ***p<0.001, ****p<0.0001. All statistical details including the number of biological or technical replicates can be found in each figure legend. Statistical analysis and data visualization were performed in GraphPad Prism 8.0 software. Graphs are presented with error bars as ± SD if not stated differently in the figure legends.

## Acknowledgements

We thank Lucie Nesvadbová, Lenka Bryjová, Pavlína Žofka Mrhálková, Lenka Doubková, and Naďa Bílá for excellent assistance. We also would like to express our gratitude to Adrienne A Boire and Jan Remsik (Memorial Sloan Kettering Cancer Center) and to Stjepan Uldrijan (Masaryk University) for their help with study models.

## Additional information

### Funding

| Funder | Grant reference number | Author |
|---|---|---|
| Czech Science Foundation | GX19-28347X | Tomasz Radaszkiewicz Vítězslav Bryja |
| Ministry of Education, Youth and Sports | LM2018127 | Zbyněk Zdráhal |
| Ministry of Education, Youth and Sports | LM2018140 | Zbyněk Zdráhal |

The funders had no role in study design, data collection and interpretation, or the decision to submit the work for publication.

### Author contributions

Tomasz Radaszkiewicz, Conceptualization, Formal analysis, Investigation, Methodology, Validation, Visualization, Writing - original draft, Writing - review and editing; Michaela Nosková, Katarzyna Anna Radaszkiewicz, Markéta Picková, Investigation, Methodology; Kristína Gömöryová, Data curation, Formal analysis, Investigation, Methodology, Software, Visualization; Olga Vondálová Blanářová, Investigation, Methodology, Visualization; Ráchel Víchová, Tomáš Gybeľ, Lucia Demková, Investigation; Karol Kaiser, Investigation, Resources; Lucia Kučerová, Zbyněk Zdráhal, Resources, Supervision; Tomáš Bárta, Dr. Tomáš Bárta performed experiments applied in the revised submission. Authors agree with their inclusion and place in the author list, Investigation; David Potěšil, Data curation, Investigation, Methodology, Supervision; Karel Souček, Investigation, Methodology, Supervision; Vítězslav Bryja, Conceptualization, Funding acquisition, Supervision, Validation, Writing - original draft, Writing - review and editing

### Author ORCIDs

Tomasz Radaszkiewicz http://orcid.org/0000-0003-4850-9933
Olga Vondálová Blanářová http://orcid.org/0000-0002-5998-5348
Katarzyna Anna Radaszkiewicz http://orcid.org/0000-0001-9604-3950
Karol Kaiser http://orcid.org/0000-0003-4705-2003
Vítězslav Bryja http://orcid.org/0000-0002-9136-5085

### Ethics

Animal experiments were approved by the Academy of Sciences of the Czech Republic (AVCR 85/2018), supervised by the local ethical committee, and performed by certified individuals.

### Decision letter and Author response

Decision letter https://doi.org/10.7554/eLife.65759.sa1
Author response https://doi.org/10.7554/eLife.65759.sa2

---

## Additional files

### Supplementary files

• Transparent reporting form
• Supplementary file 1. Sequencing of the CRISPR/Cas9 derived cell lines.

### Data availability

All data generated or analysed during this study are included in the manuscript and supporting files. Source data files are also provided.

The following previously published datasets were used:

| Author(s) | Year | Dataset title | Dataset URL | Database and Identifier |
|---|---|---|---|---|
| Spit M, Fenderico N, Jordens I, Radaszkiewicz T, Lindeboom RG, Bugter JM, Cristobal A, Ootes L, van Osch M, Janssen E, Boonekamp KE, Hanakova K, Potesil D, Zdrahal Z, Boj SF, Medema JP, Bryja V, Koo BK, Vermeulen M, Maurice MM | 2020 | RNF43 truncations trap CK1 to drive niche-independent self-renewal in cancer | http://proteomecentral.proteomexchange.org/cgi/GetDataset?ID=PXD020478 | ProteomeXchange, PXD020478 |

---

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

# Appendix 1

## Appendix 1—key resources table

| Reagent type (species) or resource | Designation | Source or reference | Identifiers | Additional information |
|---|---|---|---|---|
| Strain, strain background (*Mus musculus*) | NOD-Rag1$^{null}$ IL2rg$^{null}$ | Jackson Laboratory | RRID:BCBC_1261 | 9 -week-old males |
| Cell line (*Homo sapiens*) | T-REx 293 | Thermo Fisher Scientific | R71007; RRID:CVCL_D585 | For TetON system using pcDNA4-TO backbone |
| Cell line (*Homo sapiens*) | T-REx 293 RNF43 TetON | This publication | | Cells inducibly overexpressing RNF43 |
| Cell line (*Homo sapiens*) | T-REx 293 RNF43 Mut1 TetON | This publication | | Cells inducibly overexpressing inactive RNF43 |
| Cell line (*Homo sapiens*) | T-REx 293 RNF43/ZNRF3 dKO | This publication | | Cells lacking RNF43/ZNRF3; CRISPR/Cas9 |
| Cell line (*Homo sapiens*) | T-REx 293 RNF43/ZNRF3 dKO RNF43 TetON | This publication | | RNF43/ZNRF3 dKO inducibly overexpressing RNF43 |
| Cell line (*Homo sapiens*) | T-REx 293 DVL1/2/3 tKO | *Paclíková et al., 2017* | | Cells lacking all DVL isoforms; CRISPR/Cas9 |
| Cell line (*Homo sapiens*) | T-REx 293 WNT5A/B KO | This publication | | Cells lacking WNT5A/B isoforms; CRISPR/Cas9 |
| Cell line (*Homo sapiens*) | T-REx 293 ROR1 KO | This publication | | Cells lacking ROR1; CRISPR/Cas9 |
| Cell line (*Homo sapiens*) | A375 (amelanotic malignant melanoma) | *Kucerova et al., 2014* | RRID:CVCL_0132 | BRAF V600E; GFP constitutive expression |
| Cell line (*Homo sapiens*) | A375 RNF43/ZNRF3 dKO | This publication | | Cells lacking RNF43/ZNRF3; CRISPR/Cas9 |
| Cell line (*Homo sapiens*) | A375+ RNF43 | This publication | | Stable overexpression of RNF43 |
| Cell line (*Homo sapiens*) | A375 RNF43 TetON | This publication | | Cells inducibly overexpressing RNF43 |
| Cell line (*Homo sapiens*) | A375 TetON ctrl | This publication | | Control cells for A375 RNF43 TetON |
| Cell line (*Homo sapiens*) | A375 IV (amelanotic malignant melanoma) | *Kucerova et al., 2014* | | BRAF V600E; GFP constitutive expression |
| Cell line (*Homo sapiens*) | A375 IV RNF43/ZNRF3 dKO | This publication | | Cells lacking RNF43/ZNRF3; CRISPR/Cas9 |
| Cell line (*Homo sapiens*) | A375 IV + RNF43 | This publication | | Stable overexpression of RNF43 |
| Cell line (*Homo sapiens*) | A2058 (metastatic melanoma) | ECACC | 91100402; RRID:CVCL_1059 | BRAF V600E |
| Cell line (*Homo sapiens*) | A2058 RNF43 TetON | This publication | | Cells inducibly overexpressing RNF43 |
| Cell line (*Homo sapiens*) | MelJuso | Laboratory of Stjepan Uldrijan | RRID:CVCL_1403 | NRAS$^{Q61L/WT}$ HRAS$^{G13D/G13D}$ |
| Cell line (*Homo sapiens*) | MelJuso RNF43 TetON | This publication | | Cells inducibly overexpressing RNF43 |
| Peptide, recombinant protein | Recombinant human R-Sponidin-1 | PeproTech | 120-38 | Final concentration 50 ng/ml |
| Peptide, recombinant protein | Recombinant human WNT3A | R&D Systems | 5036-WN | Range 40–80 ng/ml |
| Peptide, recombinant protein | Recombinant human WNT5A | R&D Systems | 645-WN | Range 40–80 ng/ml |

*Appendix 1 Continued on next page*

*Appendix 1 Continued*

| Reagent type (species) or resource | Designation | Source or reference | Identifiers | Additional information |
| --- | --- | --- | --- | --- |
| Chemical compound, drug | LGK-974, Porcupine inhibitor | PeproTech | 1241454 | 1 µM |
| Chemical compound, drug | C-59, Porcupine inhibitor | Abcam | ab142216 | 0.5 µM |
| Chemical compound, drug | Dansylcadaverine, inhibitor of clathrin-dependent endocytosis | Sigma-Aldrich | D4008 | 50 µM |
| Chemical compound, drug | Chloroquine, inhibitor of lysosomal hydrolases | Sigma-Aldrich | C662 | 10 µM |
| Chemical compound, drug | MG-132, proteasome inhibitor | Sigma-Aldrich | C2211 | 10 µM |
| Chemical compound, drug | Vemurafenib BRAF V600E inhibitor | MedChem Express | HY-12057 | Up to 5 µM |
| Chemical compound, drug | Cobimetinib, Mek1 inhibitor | MedChem Express | HY-13064 | Up to 0.5 µM |
| Antibody | β-actin (rabbit monoclonal) | Cell Signaling Technology | CS-4970; RRID:AB_2223172 | WB (1:3000) |
| Antibody | DVL-2 (rabbit polyclonal) | Cell Signaling Technology *Mentink et al., 2018* | CS-3216; RRID:AB_2093338 | WB (1:1000) |
| Antibody | DVL-3 (rabbit polyclonal) | Cell Signaling Technology *Mentink et al., 2018* | CS-3218; RRID:AB_10694060 | WB (1:1000) |
| Antibody | DVL-3 (rabbit monoclonal) | Santa Cruz Biotechnology *Kaiser et al., 2019* | SC-8027; RRID:AB_627434 | WB (1:1000) |
| Antibody | Phospho-p44/42 MAPK (Erk1/2) (Thr202/Tyr204) (rabbit polyclonal) | Cell Signaling Technology *Radaszkiewicz et al., 2020* | CS-9101; RRID:AB_331646 | WB (1:1000) |
| Antibody | Total MAPK (Erk1/2) (rabbit monoclonal) | Cell Signaling Technology *Radaszkiewicz and Bryja, 2020* | CS-4695; RRID:AB_390779 | WB (1:1000) |
| Antibody | ROR1 (rabbit polyclonal) | Kind gift from *Ho et al., 2012* | | WB (1:3000) |
| Antibody | ROR2 (mouse monoclonal) | Santa Cruz Biotechnology *Ozeki et al., 2016* | sc-374174; RRID:AB_10989358 | WB (1:1000) |
| Antibody | WNT5A (rat monoclonal) | R&D Systems *Kaiser et al., 2019* | MAB645; RRID:AB_10571221 | WB (1:500) |
| Antibody | WNT11 (rabbit polyclonal) | LifeSpan BioSciences *Kotrbová et al., 2020* | LS-C185754 | WB (1:500) |
| Antibody | VANGL2 2 G4 (rat monoclonal) | Merck *Mentink et al., 2018* | MABN750; RRID:AB_2721170 | WB (1:500) |
| Antibody | MITF (rabbit monoclonal) | Cell Signaling Technology *Lavelle et al., 2020* | CS-12590; RRID:AB_2616024 | WB (1:1000) |
| Antibody | HA-11 (mouse monoclonal) | Covance *Paclíková et al., 2017* | MMS-101R; RRID:AB_291262 | WB (1:2000); IF (1:500); IP (1 µg) |

*Appendix 1 Continued on next page*

*Appendix 1 Continued*

| Reagent type (species) or resource | Designation | Source or reference | Identifiers | Additional information |
|---|---|---|---|---|
| Antibody | HA (rabbit polyclonal) | Abcam *Paclíková et al., 2017* | ab9110; RRID:AB_307019 | WB (1:2000); IF (1:500); IP (1 µg); FC (1:1000) |
| Antibody | c-Myc (9E10) (mouse monoclonal) | Santa Cruz Biotechnology *Hanáková et al., 2019* | sc-40; RRID:AB_2857941 | WB (1:500); IF (1:250); IP (1 µg) |
| Antibody | GFP 3 H9 (rat monoclonal) | Chromotek *Harnoš et al., 2019* | 3 H9 | WB (1:2000); IP (1 µg) |
| Antibody | GFP (rabbit polyclonal) | Fitzgerald *Hanáková et al., 2019* | 20R-GR-011; RRID:AB_1286217 | WB (1:2000); IP (1 µg) |
| Antibody | FLAG M2 (mouse monoclonal) | Sigma-Aldrich *Paclíková et al., 2017* | F3165; RRID:AB_259529 | WB (1:2000), IF (1:500) |
| Antibody | FLAG (rabbit polyclonal) | Sigma *Paclíková et al., 2017* | F7425; RRID:AB_439687 | WB (1:2000); IF (1:500) |
| Antibody | V5 (mouse monoclonal) | Thermo Fisher Scientific *Kaiser et al., 2019* | R96025; RRID:AB_159313 | WB (1:1000), IF (1:1000); IP (1 µg) |
| Antibody | Cortactin (mouse monoclonal) | Santa Cruz Biotechnology *Weeber et al., 2019* | sc-55579; RRID:AB_831187 | IF (1:250) |
| Antibody | Golgin-97 (mouse monoclonal) | Invitrogen | A-21270; RRID:AB_221447 | IF (1:500) |
| Antibody | a-mouse IgG HRP (goat polyclonal) | Sigma-Aldrich | A4416; RRID:AB_258167 | WB (1:4000) |
| Antibody | a-rabbit IgG HRP (goat polyclonal) | Sigma-Aldrich | A0545; RRID:AB_257896 | WB (1:4000) |
| Antibody | a-rat IgG HRP (goat polyclonal) | Sigma-Aldrich | A9037; RRID:AB_258429 | WB (1:4000) |
| Antibody | Streptavidin-HRP conjugate | Abcam | ab7403 | WB (1:4000) |
| Antibody | Ror1-APC (mouse monoclonal) | Miltenyi Biotec *Kotašková et al., 2016* | 30-119-860 | FC (1:25) |
| Antibody | a-mouse Alexa Fluor 488 (goat polyclonal) and 568 (donkey polyclonal) | Thermo Fisher Scientific | A-11001 (RRID:AB_2534069) and A10037 (RRID:AB_2534013) | IF (1:600) |
| Antibody | a-rabbit Alexa Fluor 488 (donkey polyclonal) and 568 (goat polyclonal) | Thermo Fisher Scientific | A21206 (RRID:AB_2535792) and A11011 (RRID:AB_143157) | IF (1:600) |
| Antibody | Streptavidin, Alexa Fluor 488 conjugate | Thermo Fisher Scientific | S-32354 | IF (1:600) |
| Antibody | Phalloidine Alexa Fluor 594 | Thermo Fisher Scientific | A12381 | IF (1:600) |
| Antibody | Phalloidine 4 Alexa Fluor 488 | Thermo Fisher Scientific | A12379 | IF (1:600) |
| Recombinant DNA reagent | pcDNA4-TO-RNF43-2xHA-2xFLAG | Kindly gifted by Bon-Kyoung Koo (*Koo et al., 2012*) | | Inducible expression of RNF43; backbone for cloning |
| Recombinant DNA reagent | pcDNA4-TO-RNF43Mut1-2xHA-2xFLAG | Kindly gifted by Bon-Kyoung Koo (*Koo et al., 2012*) | | Inducible expression of inactive RNF43-HA-FLAG |

*Appendix 1 Continued on next page*

*Appendix 1 Continued*

| Reagent type (species) or resource | Designation | Source or reference | Identifiers | Additional information |
| --- | --- | --- | --- | --- |
| Recombinant DNA reagent | pcDNA4-TO-RNF43-BirA*-HA | This publication | | Inducible expression of RNF43 with BirA* and HA tags |
| Recombinant DNA reagent | pcDNA3-RNF43-HA | This publication | | Expression of RNF43-HA |
| Recombinant DNA reagent | pCW57-RNF43 | This publication | | Lentiviral plasmid allowing inducible expression of HA/FLAG tagged RNF43 |
| Recombinant DNA reagent | myc-Vangl1, GFP-Vangl2, GFP-Vangl2ΔN, GFP-Vangl2ΔC, GFP-Vangl2ΔNΔC | *Belotti et al., 2012* | | Expression of VANGL1 and VANGL2 and their variants |
| Recombinant DNA reagent | pLAMP1-mCherry | Addgene #45147 *Van Engelenburg and Palmer, 2010* | RRID:Addgene_45147 | Expression of lysosomes marker |
| Recombinant DNA reagent | pEGFP-C1-Rab5a | *Chen et al., 2009* | | Expression of early endosomes marker |
| Recombinant DNA reagent | GFP-rab11 WT | Addgene #12674 *Choudhury et al., 2002* | RRID:Addgene_12674 | Expression of recycling endosomes marker |
| Recombinant DNA reagent | His-ubiquitin | *Tauriello et al., 2010* | | Tagged ubiquitin for His-Ub pulldown assay |
| Recombinant DNA reagent | pcDNA3-Flag-mDvl1 | *Tauriello et al., 2010* | | Expression of Dvl1 with Flag tag |
| Recombinant DNA reagent | pCMV5-3xFlag Dvl2 | Addgene #24802 *Narimatsu et al., 2009* | RRID:Addgene_24802 | Expression of DVL2 with Flag tag |
| Recombinant DNA reagent | pCDNA3.1-Flag-hDvl3 | *Angers et al., 2006* | | Expression of DVL3 with Flag tag |
| Recombinant DNA reagent | pcDNA3.1-hROR1-V5-His | gifted by Kateřina Tmějová | | Expression of ROR1 with V5 tag |
| Recombinant DNA reagent | pcDNA3-Ror2-Flag; pcDNA3-Ror2-dCRD-FLAG | *Sammar et al., 2004* | | Expression of ROR2 with FLAG tag and its mutant lacking CRD domain |
| Recombinant DNA reagent | pRRL2_ROR1ΔCYTO and pRRL2_ROR1ΔTail | *Gentile et al., 2011* | | Expression of ROR1 and its truncated versions |
| Recombinant DNA reagent | hCas9 | Addgene #41815 *Mali et al., 2013* | RRID:Addgene_41815 | Humanized Cas9 |
| Recombinant DNA reagent | gRNA_GFP-T1 | Addgene #41819 *Mali et al., 2013* | RRID:Addgene_41819 | gRNA expression plasmid |
| Recombinant DNA reagent | PiggyBack-Hygro; Transposase | Gifted by Bon-Kyoung Koo | RRID:Addgene_64324 | PiggyBack transposase system for stable cell lines generation |
| Recombinant DNA reagent | pU6-(BbsI)CBh-Cas9-T2A-mCherry | Addgene #64324 *Chu et al., 2015* | RRID:Addgene_64324 | All-in-1 Cas9 plasmid |
| Recombinant DNA reagent | pSpCas9(BB)-2A-GFP (PX458) | Addgene #48138 *Ran et al., 2013* | RRID:Addgene_48138 | All-in-1 Cas9 plasmid |
| Sequence-based reagent | *RNF43* gRNA | This publication | gRNA | TGAGTTCCATCGTAACTGTGTGG |
| Sequence-based reagent | *ZNRF3* gRNA | This publication | gRNA | AGACCCGCTCAAGAGGCCGGTGG |
| Sequence-based reagent | *WNT5A* gRNA | This publication | gRNA | AGTATCAATTCCGACATCGAAGG |
| Sequence-based reagent | *ROR1* gRNA | This publication | gRNA | CCATCTATGGCTCTCGGCTGCGG |
| Sequence-based reagent | *RNF43* gRNA | This publication | gRNA | AGTTACGATGGAACTCATGG |
| Sequence-based reagent | *ZNRF3* gRNA | This publication | gRNA | CTCCAGACAGATGGCACAGTCGG |

*Appendix 1 Continued on next page*

*Appendix 1 Continued*

| Reagent type (species) or resource | Designation | Source or reference | Identifiers | Additional information |
|---|---|---|---|---|
| Sequence-based reagent | B2M_F | This publication | qPCR primer | CACCCCCACTGAAAAAGATG |
| Sequence-based reagent | B2M_R | This publication | qPCR primer | ATATTAAAAAGCAAGCAAGCAGAA |
| Sequence-based reagent | GAPDH_F | This publication | qPCR primer | GACAGTCAGCCGCATCTTCT |
| Sequence-based reagent | GAPDH_R | This publication | qPCR primer | TTAAAAGCAGCCCTGGTGAC |
| Sequence-based reagent | HSPCB_F | This publication | qPCR primer | TCTGGGTATCGGAAAGCAAGCC |
| Sequence-based reagent | HSPCB_R | This publication | qPCR primer | GTGCACTTCCTCAGGCATCTTG |
| Sequence-based reagent | RPS13_F | This publication | qPCR primer | CGAAAGCATCTTGAGAGGAACA |
| Sequence-based reagent | RPS13_R | This publication | qPCR primer | TCGAGCCAAACGGTGAATC |
| Sequence-based reagent | RNF43_F | This publication | qPCR primer | TTTCCTGCCTCCATGAGTTC |
| Sequence-based reagent | RNF43_R | This publication | qPCR primer | CAGGGACTGGGAAAATGAATC |
| Sequence-based reagent | ZNRF3_F | This publication | qPCR primer | GCTTTCTTCGTCGTGGTCTC |
| Sequence-based reagent | ZNRF3_R | This publication | qPCR primer | GCCTGTTCATGGAATTCTGAC |
| Sequence-based reagent | DVL2_F | This publication | qPCR primer | TCCTTCCACCCTAATGTGTCCA |
| Sequence-based reagent | DVL2_R | This publication | qPCR primer | CATGCTCACTGCTGTCTCTCCT |
| Sequence-based reagent | DVL3_F | This publication | qPCR primer | ACCTTGGCGGACTTTAAGGG |
| Sequence-based reagent | DVL3_R | This publication | qPCR primer | TCACCACTCCGAAATCGTCG |
| Sequence-based reagent | WNT5A_F | This publication | qPCR primer | GCAGCACTGTGGATAACACCTCTG |
| Sequence-based reagent | WNT5A_R | This publication | qPCR primer | AACTCCTTGGCAAAGCGGTAGCC |
| Sequence-based reagent | ROR1_F | This publication | qPCR primer | TCTCGGCTGCGGATTAGAAAC |
| Sequence-based reagent | ROR1_R | This publication | qPCR primer | TCCAGTGGAAGAAACCACCTC |
| Sequence-based reagent | ROR2_F | This publication | qPCR primer | GTGCGGTGGCTAAAGAATGAT |
| Sequence-based reagent | ROR2_F | This publication | qPCR primer | ATTCGCAGTCGTGAACCATATT |
| Sequence-based reagent | MITF_F | *Su et al., 2020* | qPCR primer | TGCCCAGGCATGAACACAC |
| Sequence-based reagent | MITF_R | *Su et al., 2020* | qPCR primer | GGGAAAAATACACGCTGTGAG |

WB: western blot; IF:immunofluorescence; IP: immunoprecipitation.

