## [Decision Letter]

**Acceptance summary:**

This study will be of interest to scientists studying pathways involved in cancer metastasis, as it reveals a novel regulatory mechanism involved in cancer cell invasion. Here, the authors have shown that RNF43, a ubiquitin ligase, can regulate non-canonical Wnt signaling through influencing turnover of ROR1 and ROR2, co-receptors found on the surface of cells involved in Wnt signaling. Expression of RNF43 in melanoma cell lines blocks their invasive properties, prevents resistance to BRAF and MEK inhibitors, and improves response to targeted therapies in vivo. Therefore, RNF43 is a newly discovered negative regulator of WNT5A-mediated biological responses.

**Decision letter after peer review:**

Thank you for submitting your article "RNF43 inhibits WNT5A driven signaling and suppresses melanoma invasion" for consideration by *eLife*. Your article has been reviewed by 3 peer reviewers, and the evaluation has been overseen by a Reviewing Editor and Erica Golemis as the Senior Editor. The reviewers have opted to remain anonymous.

The reviewers have discussed with each other and have agreed that some major issues that would need to be addressed before publication in *eLife*.

Major Points:

1. in vivo experiments would be needed to show the effect of RNF43 in the context of resistance to vemurafenib. If these cannot be performed, amend the title of the paper to better reflect the experiments performed, as at the moment it refers to melanoma invasion.

2. The number of cell lines used is not sufficient. All reviewers noted that, in order to rule out cell-line specific effects, the effects of RNF43 on the WNT5A pathway and their functional relevance should be tested in at least one additional melanoma cell line. One reviewer noted that "The findings in melanoma are weakened by the fact that only 1 pair of isogenic cell lines is used. An additional distinct cell line would greatly strengthen the paper findings and impact." It is essential to perform these experiments in additional melanoma cell models.

3. Mechanistic function of RNF43. Please provide data to clarify the following points:

– The authors mention that RNF43 ubiquitination of VANGL2 leads to downregulation of ROR1. Does it also lead to a downregulation of ROR2, in melanoma cells?

– What happens to WNT5A expression in the RNF43-KO cell lines (A375 WT and A375 WT VR)?

– What causes a loss of expression of RNF43 during melanoma progression?

4. Figure 5A. How do the authors address the lack of effect, on migration, in A375 IV overexpressing RNF43? Also, what statistical test was used to compare the graphs? There is one line comparing 48 h in A375 WT +/- RNF43 showing a difference, but it is difficult to tell if there is a difference between A375 IV without and with overexpression of RNF43.

– Figure 5B and 5C. What is the explanation for the lack of effect of RNF43 KO?

– Figure 6F. How do the authors explain the increase in sensitivity to vemurafenib following RNF43 KO?

5. Due to the clinical relevance of combined treatment, it would be more appropriate to test the effects of RNF43 in the context of BRAF and MEK inhibition, rather than BRAF inhibition alone. Please add these experiments.

---

## [Author Response]

The reviewers have discussed with each other and have agreed that some major issues that would need to be addressed before publication in eLife.

We would like to thank all reviewers for their valuable comments and useful suggestions, which helped us to improve our manuscript. We have expanded our work in all directions suggested by the editor. Namely, (i) we have introduced new cell lines and also new and more robust invasion/chemotaxis assays, (ii) the work on targeted therapies has been improved by introduction of MEK inhibitors and their combination with BRAF V600E inhibition, and finally, (iii) we have repeated the key findings (the effect on Vemurafenib resistance) in the in vivo xenograft experiment. We also decided to modify the title of our study to reflect better the actual content of our publication:

“RNF43 inhibits WNT5A driven signaling, suppresses melanoma invasion and resistance to the targeted therapy”.

Major Points:1. In vivo experiments would be needed to show the effect of RNF43 in the context of resistance to vemurafenib. If these cannot be performed, amend the title of the paper to better reflect the experiments performed, as at the moment it refers to melanoma invasion.

To address this issue, we have set up an in vivo assay where A375 were administered intradermally into immunodeficient NOD-Rag1^null^ IL2rg^null^ mice. We have used a panel of A375 derivatives representing *RNF43*-low, *RNF43*-medium and *RNF43*-high conditions and questioned if *RNF43* expression level affects tumor formation and response to Vemurafenib. The sets of in vivo experiments are presented as the novel Figure 7 and associated supplementary figure. Importantly, mouse xenograft data confirm our concept defined in the in vitro conditions and strengthen our study as such.

2. The number of cell lines used is not sufficient. All reviewers noted that, in order to rule out cell-line specific effects, the effects of RNF43 on the WNT5A pathway and their functional relevance should be tested in at least one additional melanoma cell line. One reviewer noted that "The findings in melanoma are weakened by the fact that only 1 pair of isogenic cell lines is used. An additional distinct cell line would greatly strengthen the paper findings and impact." It is essential to perform these experiments in additional melanoma cell models.

Thank you for this suggestion. As part of work on the revision we have expanded our panel of cell lines and included two additional melanoma cell lines: A2058 (BRAF V600E) and MelJuso (NRAS^Q61L/WT^ HRAS^G13D/G13D^). These cell lines were genetically modified to inducibly express RNF43. Both lines were analyzed for their ability to respond to WNT5A (Figure 4J and Figure 4 figure supplement 3A). A2058 were also tested in the functional assays – wound healing (Figure 5E), collagen I hydrogel invasion assay (Figure 5H) and acquisition of resistance to Vemurafenib, Cobimetinib and combined therapy (Figure 6H-I and Figure 6 figure supplement 1C-F). We could confirm our earlier T-REx 293 and A375-based observations that RNF43 inhibits WNT5A signaling, melanoma invasion and acquired resistance.

3. Mechanistic function of RNF43. Please provide data to clarify the following points:– The authors mention that RNF43 ubiquitination of VANGL2 leads to downregulation of ROR1. Does it also lead to a downregulation of ROR2, in melanoma cells?

We understand that by “downregulation of ROR1” reviewer meant “downregulation of ROR1 cell surface level” and “downregulation of ROR1 phosphorylation” that we show in the manuscript. Based on known similarities between ROR1 and ROR2 we expected that ROR2 will behave similarly as ROR1 and updated Figure 4 in this respect. Indeed, electrophoretic mobility shift of ROR2 is blocked in A2058 RNF43 TetON (Figure 4J) and A375 RNF43 TetON (Figure 4 figure supplement 3B) cells. Thus, we concluded that RNF43 decreases phosphorylation of ROR1 as well as ROR2.

– What happens to WNT5A expression in the RNF43-KO cell lines (A375 WT and A375 WT VR)?

To answer this issue, we showed in the Figure 6 data analyzing *WNT5A* expression. We can conclude that during the process for resistance acquisition *WNT5A* expression is increased. It is true in Vemurafenib resistant A375 cells (Figure 6E) as well as A2058 cells resistant to Vemurafenib, Cobimetinib and their FDA-approved combination (Figure 6K). The process is accompanied by decreased *RNF43* (Figure 6J) and *ZNRF3* (Figure 6 supplement 1D) expression compared to the parental cells. We also noticed possible changes in the melanoma phenotypes descried by *WNT5A*/*MITF*, which is now part of the *Discussion* part.

– What causes a loss of expression of RNF43 during melanoma progression?

We were unable to answer this question experimentally. However, in the revised version of the manuscript, we have updated *Discussion* to speculate on this issue. Specifically, we propose that the loss of RNF43 is a marker of phenotypic changes associated with *MITF*-low/*WNT5A*-high phenotype (Figure 6I and 6K). *RNF43*, similarly to *MITF*, is a target gene of the canonical Wnt pathway main component – β-catenin (Takahashi et al., 2014). High expression of *WNT5A* and low transcriptional activity of β-catenin are factors describing invasive and therapy resistant cells, as discussed previously (Ahn et al., 2017; Arozarena and Wellbrock, 2019; Eichhoff et al., 2011; Kaur et al., 2016; Prasad et al., 2015).

4. Figure 5A. How do the authors address the lack of effect, on migration, in A375 IV overexpressing RNF43? Also, what statistical test was used to compare the graphs? There is one line comparing 48 h in A375 WT +/- RNF43 showing a difference, but it is difficult to tell if there is a difference between A375 IV without and with overexpression of RNF43.

In the revised manuscript we present all 2D migration (wound healing assay) data in the same format; statistically compared after 48 h of migration (Figure 5A-E) for all four experimental models – A375 and A375 IV stable cell lines, and A375 and A2058 RNF43 TetON cell lines. Data presented in this format clearly show the effects of increased RNF43 on wound healing in all cases. In addition, we present novel data analyzing directional 3D invasion with very clear effects of RNF43 overexpression both in A375 and A2058 cells. Unpaired two-tailed t-test has been used to demonstrate the significance of the RNF43 effect.

– Figure 5B and 5C. What is the explanation for the lack of effect of RNF43 KO?

Lack of *RNF43*/*ZNRF3* dKO effect in Matrigel invasion, invadopodia formation and gelatin degradation in both A375 and A375 IV lines suggests that already wild type cells have negligible activity of RNF43. Indeed, *RNF43* expression in these lines is very low (see Figure 4 figure supplement 2B). As such the deletion of *RNF43*/*ZNRF3* does not further improve their Wnt-driven capacity for migration and invasion that is already very high.

– Figure 6F. How do the authors explain the increase in sensitivity to vemurafenib following RNF43 KO?

Both A375 and IV *RNF43*/*ZNRF3* dKO survived Vermurafenib selection, although *RNF43*/*ZNRF3* dKO cells seem to be in lower numbers (new Figure 6G). We do not know what mechanism is behind this phenotype, but we think that it can be a consequence of decreased proliferation rate of *RNF43*/*ZNRF3* dKO cells. *RNF43*/*ZNRF3* dKO cells have elevated WNT5A pathway activity (i.e. Figure 3B and Figure 4H, I) and increased *WNT5A* expression accompanies the acquisition of resistance to melanoma targeted therapies (Figure 6E and K). It was shown before that high *WNT5A* expression causes a senescence-like phenotype of melanoma after Vemurafenib (Webster et al., 2015). This explanation has been added to the main manuscript text.

5. Due to the clinical relevance of combined treatment, it would be more appropriate to test the effects of RNF43 in the context of BRAF and MEK inhibition, rather than BRAF inhibition alone. Please add these experiments.

In response to the reviewer´s comment we have added this experiment to our study. We tested the cellular responses to the short term and chronic exposure to MEK inhibitor Cobimetinib and its FDA-approved combination with Vemurafenib using A2058 cells (Figure 6H – L and Figure 6 supplement 1C-F). As expected, RNF43 overexpressing cells did not survive selection (Figure 6L). Interestingly, cells resistant to MEK inhibition alone and to combo BRAF V600Ei+MEKi have decreased *RNF43* and *ZNRF3* expression and increased *WNT5A*. The new data and their discussion are now present in Figure 6 and accompanying text.

References

Ahn A, Chatterjee A, Eccles MR. 2017. The slow cycling phenotype: A growing problem for treatment resistance in melanoma. Mol Cancer Ther. doi:10.1158/1535-7163.MCT-16-0535

Arozarena I, Wellbrock C. 2019. Phenotype plasticity as enabler of melanoma progression and therapy resistance. Nat Rev Cancer. doi:10.1038/s41568-019-0154-4

Eichhoff OM, Weeraratna A, Zipser MC, Denat L, Widmer DS, Xu M, Kriegl L, Kirchner T, Larue L, Dummer R, Hoek KS. 2011. Differential LEF1 and TCF4 expression is involved in melanoma cell phenotype switching 24:631–642. doi:10.1111/j.1755-148X.2011.00871.x

Feng D, Wang J, Yang W, Li J, Lin X, Zha F, Wang X, Ma L, Choi NT, Mii Y, Takada S, Huen MSY, Guo Y, Zhang L, Gao B. 2021. Regulation of Wnt/PCP signaling through p97/VCP-KBTBD7–mediated Vangl ubiquitination and endoplasmic reticulum–associated degradation. Sci Adv 7:eabg2099. doi:10.1126/sciadv.abg2099

Guo Y, Zanetti G, Schekman R. 2013. A novel GTP-binding protein-adaptor protein complex responsible for export of Vangl2 from the trans Golgi network. eLife 2013:160. doi:10.7554/eLife.00160

Kaur A, Webster MR, Weeraratna AT. 2016. In the Wnt-er of life: Wnt signalling in melanoma and ageing. Br J Cancer 115:1273–1279. doi:10.1038/bjc.2016.332

Prasad C, Mohapatra P, Andersson T. 2015. Therapy for BRAFi-Resistant Melanomas: Is WNT5A the Answer? Cancers (Basel) 7:1900–1924. doi:10.3390/cancers7030868

Takahashi N, Yamaguchi K, Ikenoue T, Fujii T, Furukawa Y. 2014. Identification of two wnt-responsive elements in the intron of RING finger protein 43 (RNF43) gene. PLoS One. doi:10.1371/journal.pone.0086582

Tower-Gilchrist C, Zlatic SA, Yu D, Chang Q, Wu H, Lin X, Faundez V, Chen P. 2019. Adaptor protein-3 complex is required for Vangl2 trafficking and planar cell polarity of the inner ear. Mol Biol Cell 30:2422–2434. doi:10.1091/mbc.E16-08-0592

Webster MR, Xu M, Kinzler KA, Kaur A, Appleton J, O’Connell MP, Marchbank K, Valiga A, Dang VM, Perego M, Zhang G, Slipicevic A, Keeney F, Lehrmann E, Wood W, Becker KG, Kossenkov A V., Frederick DT, Flaherty KT, Xu X, Herlyn M, Murphy ME, Weeraratna AT. 2015. Wnt5A promotes an adaptive, senescent-like stress response, while continuing to drive invasion in melanoma cells 28:184–195. doi:10.1111/pcmr.12330

Yang W, Garrett L, Feng D, Elliott G, Liu X, Wang N, Wong YM, Choi NT, Yang Y, Gao B. 2017. Wnt-induced Vangl2 phosphorylation is dose-dependently required for planar cell polarity in mammalian development. Cell Res 27:1466–1484. doi:10.1038/cr.2017.127